# INCREASING-MARGIN ADVERSARIAL (IMA) TRAINING TO IMPROVE ADVERSARIAL ROBUSTNESS OF NEURAL NETWORKS

## ABSTRACT

Deep neural networks (DNNs), including convolutional neural networks, are known to be vulnerable to adversarial attacks, which may lead to disastrous consequences in life-critical applications. Adversarial samples are usually generated by attack algorithms and can also be induced by white noises, and therefore the threats are real. In this study, we propose a novel training method, named Increasing Margin Adversarial (IMA) Training, to improve DNN robustness against adversarial noises. During training, the IMA method increases the margins of training samples by moving the decision boundaries of the DNN model far away from the training samples to improve robustness. The IMA method is evaluated on six publicly available datasets (including a COVID-19 CT image dataset) under strong 100-PGD white-box adversarial attacks, and the results show that the proposed method significantly improved classification accuracy on noisy data while keeping a relatively high accuracy on clean data. We hope our approach may facilitate the development of robust DNN applications, especially for COVID-19 diagnosis using CT images.

## 1 INTRODUCTION

Deep neural networks (DNNs), especially convolutional neural networks (CNNs), have become the first choice for automated image analysis due to its superior performance. However, recent studies have shown that DNNs are not robust to a special type of noise, called adversarial noise. Adversarial noise was first discovered by (Szegedy et al., 2013) and then explained by (Goodfellow et al., 2014). Adversarial noises can significantly affect robustness of DNNs for a wide range of image classification applications (Akhtar & Mian, 2018), such as handwritten digits recognition (Graese et al., 2016), human face recognition (Mirjalili & Ross, 2017), and even traffic sign detection (Eykholt et al., 2018). DNN-based image segmentation can also be affected by adversarial noises because image segmentation is often realized by pixel classification.

The COVID-19 pandemic has infected millions of people and caused the death of about 1 million people as of today (WHO, 2020). A large scale study in China shows that CT had higher sensitivity for the diagnosis of COVID-19 as compared with initial reverse-transcription polymerase chain reaction (RT-PCR) from swab samples (Ai et al., 2020). As reviewed in (Shi et al., 2020), many DNN models for COVID-19 diagnosis from CT images have been developed and achieved very high classification accuracy. However, none of these studies (Shi et al., 2020) considered DNN robustness against adversarial noises. We modified a Resnet-18 model (He et al., 2016) and trained it on a public COVID-19 CT image dataset (Soares et al., 2020), and then the model robustness is tested (details in section 3.3). Fig. 1 shows a CT image (denoted by $x$) of a lung that was infected by COVID-19 and correctly classified as infected. After adding a small amount of noise $\delta$ to the image $x$, the noisy image $x + \delta$ is classified as uninfected. On the test set, although the model achieved $\geq 95\%$ accuracy on original clean images, its accuracy dropped to zero on the noise level of $0.03$. This non-robust model clearly cannot be used in real clinical applications. It may be argued that adversarial noises are created by algorithms and therefore may not exist in the real world unless some bad people want to hack the system to achieve personal gain at the expense of public health. However, on the COVID-19 dataset, we found that $2.75\%$ of the noisy samples with uniform white noises on the noise level of $0.05$, can cause the DNN model to make wrong classifications, which

shows that white-noise-induced adversarial samples can exist. For medical applications, $2.75\%$ is not a negligible number, and it is worth developing methods to improve DNN adversarial robustness.

There are mainly two categories of adversarial attacks: white-box attack and black-box attack. For a white-box attack, the attacker knows everything about the DNN to be attacked. For a black-box attack, the attacker only can use the DNN as a black-box (i.e., send an input to the DNN and get an output from the DNN, not knowing anything else). From the perspective of defense, we should consider the worst-case scenario: white-box attack. People have explored many ideas to improve robustness, but many of them have been shown to be ineffective (Uesato et al., 2018; Tramer et al., 2020). A general and effective strategy is adversarial training (Goodfellow et al., 2014; Madry et al., 2017; Miyato et al., 2018; Ding et al., 2019a), and the basic idea is to use adversarial attack algorithms to generate noisy samples and add those samples to the training set, which basically is a special data augmentation strategy. Through adversarial training, the DNN model can learn from the noisy samples and become robust to noises. Adversarial training is straightforward but computationally expensive. Thus, one needs to make sure that the generated noisy samples can indeed help to improve robustness: samples with too much noise can harm performance while samples with too little noise may have no effect at all. Generative adversarial training has been directly applied to improve robustness; however, it can only defend against black-box attack (Wang & Yu, 2019).

In this paper, we propose a novel method, Increasing-Margin Adversarial (IMA) Training, to improve robustness of deep neural networks for classification tasks. Our method aims to increase margins of training samples by moving decision boundaries far away from the samples to improve robustness. We evaluated our method on six datasets with 100-PGD white-box attack and the results show that our proposed method can achieve a significant improvement in DNN robustness against adversarial noises.

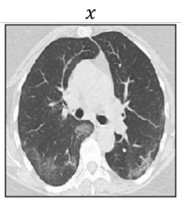 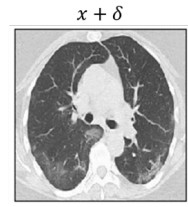 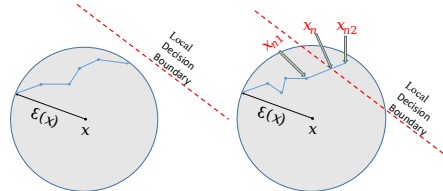

Figure 1: An example of clean and noisy images.

Figure 2: Left: case-0 in BPGD; Right: case-1 in BPGD. Please zoom in for better visualization.

## 2 METHODOLOGY

### 2.1 ADVERSARIAL ATTACK AND NEURAL NETWORK ROBUSTNESS

To evaluate the robustness of different adversarial training methods, we use projected gradient descent (PGD) (Madry et al., 2017; Kurakin et al., 2016) to generate adversarial noises, which is widely used for method evaluation (Uesato et al., 2018; Tramer et al., 2020). For the convenience of the reader, we briefly describe PGD: Let $x$ denote an input sample and $y$ be the true class label. Let $J(x)$ denote the scalar objective function of PGD, which could be the cross-entropy loss function or other classification loss functions. Let $\delta$ denote the adversarial noise, and its magnitude is $\varepsilon$ which is measured by the vector Lp norm of $\delta$, i.e., $\varepsilon = ||\delta||_p$, where $p$ is inf or 2 usually. PGD adds noises to $x$ iteratively:

$$x_{(i)} = clip\left(x_{(i-1)} + \eta \cdot h\left(J'\left(x_{(i-1)}\right)\right)\right) \tag{1}$$

where $\eta$ is step size, $J'(x) = \frac{\partial J}{\partial x}$, and $i$ is iteration index. $x_{(0)} = x + \xi$, where $\xi$ is random noise with $||\xi||_p \leq \varepsilon$. $||\ ||_p$ denotes the vector p-norm. The clip operation in Eq.(1) ensures that $||x_{(i)} - x||_p \leq \varepsilon$ (called $\varepsilon$-ball). If $x$ is an image, the clip operation also ensures that pixel values stay within the feasible range (e.g. 0 to 1). If L-inf norm is used, $h(J')$ is the sign function; and if L2 norm is used, then $h(J')$ normalizes $J'$ by its L2 norm. The total adversarial noise is $\delta = x_{(N_{PGD})} - x$, and $N_{PGD}$ is the number of iterations. $\eta$ is usually set to $\alpha \cdot \varepsilon / N_{PGD}$, and therefore the algorithm may sweep the $\varepsilon$-ball $\alpha$ times ($\alpha \geq 1$) within $N_{PGD}$ iterations. By adding

the adversarial noise $\delta$ to the input $x$, the objective function $J(x+\delta)$ will be significantly larger than $J(x)$, leading to wrong classification of the noisy sample $x+\delta$.

It is easy to understand the robustness issue through Eq.(1) and Fig. 1. Given a trained classifier, if a sample $x$ is very close to the decision boundary of the classifier, then one could add a small noise $\delta$ to $x$ such that $x+\delta$ goes across the decision boundary and is wrongly-classified. As shown in section 3, using standard training with cross-entropy loss and clean data, the decision boundaries of the trained classifier will often be very close to the samples; as a result, the accuracy can drop to zero upon very small noises. To improve robustness, vanilla adversarial training (Goodfellow et al., 2014; Kurakin et al., 2016) based on PGD can be used: given the maximum noise level $\varepsilon = \varepsilon_{max}$, generate a noisy sample $x_\varepsilon = PGD(x, y, \varepsilon)$ for every clean sample $x$ in training set; use the clean samples and noisy samples together to train the model. The loss function is given in (Goodfellow et al., 2014):

$$L_{adv} = (L_{ce}(x,y) + L_{ce}(x_\varepsilon, y))/2 \tag{2}$$

In practice, the loss is accumulated over each mini-batch. $L_{ce}$ is the cross-entropy loss. As shown in Section 3, the vanilla adversarial training is very sensitive to the user-defined noise level $\varepsilon_{max}$.

## 2.2 INCREASING-MARGIN ADVERSARIAL (IMA) TRAINING

We developed a novel method, Increasing-Margin Adversarial (IMA) training, to enhance DNN classifier robustness. As its name indicates, IMA is a type of adversarial training: it generates noisy samples and trains a DNN model on clean and noisy samples. IMA algorithms are significantly different from the other adversarial training algorithms.

IMA training includes two alternating sub-processes: Algorithm 1 to compute loss and update the DNN model, and Algorithm 2 to update margin estimation. In Algorithm 1, by minimizing the loss on clean and noisy data, the model will reach a balance between robustness and accuracy on clean data. In Algorithm 2, the sample margins are updated after each epoch. These estimated margins are used to generate noisy samples for training the model in the next epoch.

---

**Algorithm 1** compute loss and update model in an epoch

---

**Input:** : $S = (x, y)$ is the training set, containing pairs of clean sample $x$ and true label $y$. $f$ is the DNN model and $\hat{y} = f(x)$. $\mathcal{E}$ is an array containing the estimated margins of individual training samples. $\beta$ is a scalar coefficient. In a batch, $X$ contains samples and $Y$ contains class labels.

**Output:** updated model $f$ after this training epoch

    **Process:**

1: **for** Batch data $X$ and $Y$ in $S$ **do**
2:      Run model $f$ to classify the samples and divide the samples into two groups, wrongly-classified $X_0, Y_0$ and correctly-classified $X_1, Y_1$ where $X = X_0 \cup X_1$ and $Y = Y_0 \cup Y_1$
3:      Get noisy samples $X_n = BPGD(X_1, Y_1, \mathcal{E}(X_1))$ and classify them $\hat{Y}_n = f(X_n) \cdot \mathcal{E}(X_1)$ is a sub-array of $\mathcal{E}$, containing the estimated margins of samples in $X_1$. $BPGD$ is described in Algorithm 3.
4:      Compute the loss L:
5:      $L_0 = cross\_entropy(f(X_0), Y_0)$ (take sum)
6:      $L_1 = cross\_entropy(f(X_1), Y_1)$ (take sum)
7:      $L_2 = cross\_entropy(f(X_n), Y_1)$ (take sum)
8:      $L = ((1-\beta) \cdot (L_0 + L_1) + \beta \cdot L_2)/batch\_size$
9:      Back-propagate from $L$ and update the model $f$
10: **end for**

---

Our IMA method tries to generate noisy samples on decision boundaries as much as possible: adding too much noise to a training sample may lead to low accuracy on clean samples, but adding too little noise may have no effect on robustness. Since the distance between a training sample and a decision boundary is different for different training samples, the noises added to the training samples have different magnitudes (i.e., vector norms). The rationale of doing so is discussed in section 2.3. Here, the margin of a sample is the minimum distance between the sample and the decision boundaries. It is difficult to compute the exact margin for a sample in a high dimension space, and therefore we can only estimate it.

---

**Algorithm 2** update margin estimation after an epoch

---

**Input:** $S = (x, y)$ is the training set. $f$ is the DNN model and $\hat{y} = f(x)$. $\mathcal{E}$ is an array of the estimated margins of training samples. $\mathcal{E}(x)$ is the estimated margin of $x$. $\Delta_\varepsilon$ is margin expansion step size. $\varepsilon_{max}$ is the maximum noise level, i.e., the allowed maximum margin.
**Output:** updated $\mathcal{E}$
    **Process:**
  1: **for** each sample $x$ and $y$ in $S$ **do**
  2:      Classify the sample $\hat{y} = f(x)$
  3:      **if** $x$ is wrongly-classified ($\hat{y}$ is not $y$) **then**
  4:          Update $\mathcal{E}(x) = \Delta_\varepsilon$ (re-initialize)
  5:      **else if** $x$ is correctly-classified ($\hat{y}\ is\ y$) **then**
  6:          Get noisy sample $x_n = BPGD(x, y, \mathcal{E}(x))$
  7:          **if** $x_n$ is on decision boundary (not None) **then**
  8:              Update $\mathcal{E}(x) = ||x - x_n||_p$ (refine/shrink)
  9:          **else**
10:              Update $\mathcal{E}(x) = \mathcal{E}(x) + \Delta_\varepsilon$ (expand)
11:          **end if**
12:      **end if**
13: **end for**
14: Clip every element of $\mathcal{E}$ into the range of $\Delta_\varepsilon$ to $\varepsilon_{max}$
    **Note:** Every $\mathcal{E}(x)$ is initialized to be $\Delta_\varepsilon$ which can be set to $\varepsilon_{max}$ divided by the number of adversarial training epochs. The algorithm runs in mini-batches.

---

---

**Algorithm 3** (BPGD): generate noisy samples

---

**Input:** : a sample $x$ with class label $y$, the estimated margin $\varepsilon$ of $x$, and the model $f \cdot N_{PGD}$ is the number of iterations in PGD. $N_{binary}$ is the number of iterations in binary search.
**Output:** $x_n = BPGD(x, y, \varepsilon)$
    **Process:**
  1: Run the standard $PGD(x, y, \varepsilon)$ and obtain a sequence of noisy samples ($N_{PGD}$ samples)
  2: Classify those noisy samples using the model $f$
  3: **if** all of the noisy samples are correctly-classified (case-0 in Fig.2) **then**
  4:      **return** None ($x_n$ is None and will be ignored)
  5: **else**
  6:      (case-1 in Fig.2) Find $x_{n1}$ and $x_{n2}$, two adjacent samples in the sequence. $x_{n1}$ is correctly-classified and $x_{n2}$ is wrongly-classified. Thus, $x_{n1}$ and $x_{n2}$ are close to the decision boundary.
  7:      Run binary search along the straight-line segment between $x_{n1}$ and $x_{n2}$ in order to find $x_n$ on the decision boundary (Fig.3)
  8:      **return** $x_n$
  9: **end if**
    **Note:** it is unnecessary to store the whole sequence. The algorithm runs in mini-batches.

---

We developed Algorithm 3, named BPGD, to find noisy samples on decision boundaries. At the very beginning of the IMA training process, the margins of the training samples are initialized to a small number equal to margin expansion step size in Algorithm 2. Intuitively, during IMA training, the margin of a training sample keeps increasing as if a ball is expanding (the ball center is the sample $x$, and the radius is the margin $\mathcal{E}(x)$; it is called $\varepsilon$-ball), until the ball of the sample $x$ collides with the ball of another sample $x'$ in a different class. When the two balls collide and therefore a local decision boundary is formed, Algorithm 2 will prevent them from further expansion by refining/shrinking margins (Fig. 4). In section 2.3, we will show that an equilibrium state may exist (Fig. 5) under which the margins of the samples are maximized.

## 2.3 THE EQUILIBRIUM STATE

We can show that on certain conditions, an equilibrium state (Fig. 5-Left) exists under which the margins of the samples are maximized. To simplify the discussion, we assume there are three classes and three decision boundaries between classes (Fig. 5-Right). The softmax output of the neural

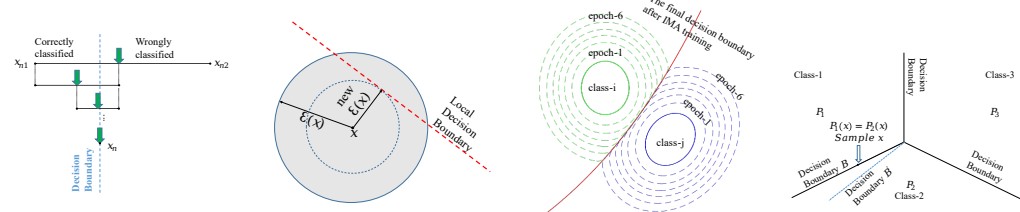

Figure 3: Binary search in BPGD.

Figure 4: Shrink Margin.

Figure 5: Left: Equilibrium State; Right: Three-class Scenario.

network model $f$ has three components: $P_1$, $P_2$ and $P_3$ corresponding to the three classes. If a point (i.e., a sample) $x$ is on the decision boundary $B_{ij}$ between class-$i$ ($c_i$) and class-$j$ ($c_j$), then $P_i(x) = P_j(x)$. The mathematical expectation of the cross-entropy loss of the noisy samples ($L_2$ in Algorithm 1) is:

$$E = \mathbf{E}_{X_n \in c_1}\left(-log\left(P_1\left(X_n\right)\right)\right) + \mathbf{E}_{X_n \in c_2}\left(-log\left(P_2\left(X_n\right)\right)\right) + \mathbf{E}_{X_n \in c_3}\left(-log\left(P_3\left(X_n\right)\right)\right) \quad (3)$$

The IMA method puts the noisy samples on the decision boundaries, thus:

$$\mathbf{E}_{X_n \in c_1}\left(-log\left(P_1\left(X_n\right)\right)\right) = \mathbf{E}_{X_n \in c_1, B_{12}}\left(-log\left(P_1\left(X_n\right)\right)\right) + \mathbf{E}_{X_n \in c_1, B_{13}}\left(-log\left(P_1\left(X_n\right)\right)\right) \quad (4)$$

$$\mathbf{E}_{X_n \in c_2}\left(-log\left(P_2\left(X_n\right)\right)\right) = \mathbf{E}_{X_n \in c_2, B_{12}}\left(-log\left(P_2\left(X_n\right)\right)\right) + \mathbf{E}_{X_n \in c_2, B_{23}}\left(-log\left(P_2\left(X_n\right)\right)\right) \quad (5)$$

$$\mathbf{E}_{X_n \in c_3}\left(-log\left(P_3\left(X_n\right)\right)\right) = \mathbf{E}_{X_n \in c_3, B_{13}}\left(-log\left(P_3\left(X_n\right)\right)\right) + \mathbf{E}_{X_n \in c_3, B_{23}}\left(-log\left(P_3\left(X_n\right)\right)\right) \quad (6)$$

If the noisy samples (random variables) $X_n \in c_i$ and $X_n \in c_j$ have the same spatial distribution on the decision boundary $B_{ij}$ between the two classes, Eq.(3) can be simplified to

$$E = E_{X_n \in B_{12}} + E_{X_n \in B_{23}} + E_{X_n \in B_{13}} \quad (7)$$

where

$$\begin{aligned} E_{X_n \in B_{ij}} &= \mathbf{E}_{X_n \in B_{ij}}\left(-log\left(P_i\left(X_n\right)\right) - log\left(P_j\left(X_n\right)\right)\right) \\ &= \mathbf{E}_{X_n \in B_{ij}}\left(-log\left(P_i\left(X_n\right)P_j\left(X_n\right)\right)\right) \\ &\geq \mathbf{E}_{X_n \in B_{ij}}\left(-log\left(\frac{P_i\left(X_n\right) + P_j\left(X_n\right)}{2}\right)^2\right) \end{aligned} \quad (8)$$

$E$ reaches the minimum when $P_i(X_n) = P_j(X_n)$.

The analysis shows that the loss of noisy samples will increase if the decision boundaries of the model $f$ change a little bit from the current state. Thus, when the loss ($L_0 + L_1$) on clean data is minimized and noisy samples are on the decision boundaries, an equilibrium is reached under the condition that noisy samples have the same spatial distribution on the decision boundaries between classes. This analysis provides the rationale that our IMA method puts the noisy samples on decision boundaries as much as possible, which is significantly different from the theory of the MMA method (Ding et al., 2019a). The dynamic process to achieve the equilibrium state is discussed in the Appendix G.

## 3 EXPERIMENTS

We applied the IMA method for six DNNs on six different datasets. For the datasets of Moons, Fashion-MNIST, SVHN, and COVID-19 CT, $\beta$ in Algorithm 1 is 0.5 because robustness and accuracy are equally important. In Algorithm 3, $N_{PGD}$ is 20, $N_{binary}$ is 10, and there is no repeat. $\alpha$ in PGD is 4. Larger $N_{PGD}$ and $N_{binary}$ may lead to better convergence of the algorithms but more computing-time. The settings and results on MNIST and CIFAR10 are reported in Appendix D. Pytorch (Paszke et al., 2017) is used for model implementation. Nvidia V100 and Titan V GPUs are used for model training and testing.

To describe the DNNs, we use "COV (a, b, c, d, e)" to represent a convolution layer with "a" input channels, "b" output channels, kernel size of "c", stride of "d" and padding of "e"; "Linear (f, g)" to denote a linear layer with input size of "f" and output size of "g"; "IN" to denote instance normalization; "LN" to denote layer normalization, and "LR" to denote leaky ReLU.

To obtain the baseline performance, each DNN was trained with cross-entropy loss on clean data, and the trained model is denoted by "ce". To evaluate the performance, we compare our method with the other three adversarial training methods, including: (1) 20-PGD ($N_{PGD}$=20 and $\alpha$=4) based vanilla adversarial training with noise level $\varepsilon$, denoted by "adv $\varepsilon$"; (2) tradeoff-inspired adversarial defense via surrogate-loss minimization, named TRADES in (Zhang et al., 2019); (3) adversarial training for direct input space margin maximization, named MMA in (Ding et al., 2019a). MMA is a state-of-the-art method, and TRADES was state-of-the-art at the time of its publication.

To make the comparison as fair as possible, the number of iterations and step size of TRADES are the same as those of the 20-PGD; and the number of PGD-iterations in MMA is 20. We used PGD to evaluate the robustness of the methods, which is widely used for method evaluation (Liao et al., 2018; Uesato et al., 2018). The number of PGD iterations is 100 to ensure a strong attack, and it is called 100-PGD. To further enhance the attack during model testing, each 100-PGD rans twice, using MarginalLoss and CrossEntropyLoss respectively, which is implemented in (Ding et al., 2019b)https://github.com/BorealisAI/advertorch. More evaluations are shown in Appendix D. Discussion about effect of coefficient $\beta$ is shown in Appendix E. In Appendix F, we show how the allowed maximum margin (i.e., $\varepsilon_{max}$) affects the performance, and how to choose its value.

## 3.1 Evaluation on Moons dataset

The Moons dataset is available in sk-learn (Pedregosa et al., 2011). There are two classes of 2D points in the dataset. The training set has 20000 samples, the validation set has 2000 samples, and the test set has 2000 samples. The neural network structure is Linear(2, 32)-LR-Linear(32, 64)-LN-LR-Linear(64, 128)-LN-LR-Linear(128, 2). The purpose of this evaluation is to visualize the decision boundaries of the models trained by different adversarial training methods. For every method, we set the number of training epochs to 30, maximum noise level to 0.3, and batch-size to 128; Adam optimizer was used with default parameters. To measure robustness on the test set, adversarial samples on different noise levels (defined by L-inf norm) are generated by the 100-PGD. The results of decision boundaries and accuracy scores are shown in Fig. 6. The accuracy scores

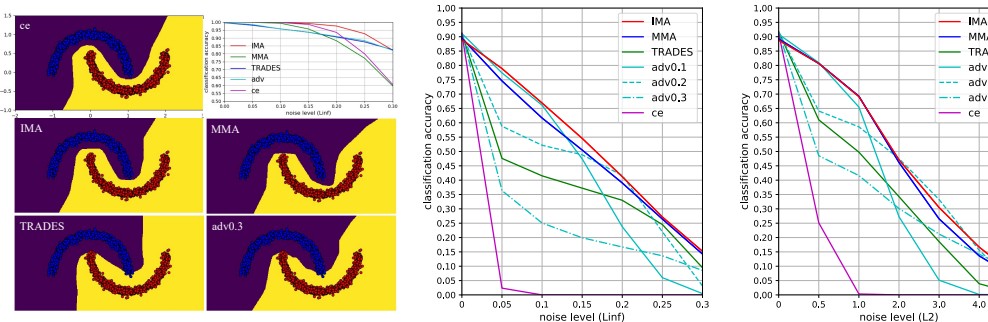

Figure 6: Results on Moons dataset.

Figure 7: Results on Fashion-MNIST (L-inf and L2 norm in 100-PGD).

are also reported in Table 1 in Appendix B. Our method IMA is better than the other methods, and its decision boundary is close to the "middle line" of the two classes. The results of TRADES and adv0.3 are very similar: the decision boundary is far away from most of the red dots. MMA is better than TRADES and adv0.3 when the noise level is less than 0.15, and then it becomes worse with larger noises. Surprisingly, "ce" (model trained with cross-entropy loss and clean data) has very good robustness when the noise level is less than 0.15.

## 3.2 Evaluation on Fashion-MNIST and SVHN datasets

We used Fashion-MNIST dataset instead of MNIST which is too easy. The dataset contains 28×28 grayscale images in 10 classes (shirts, shoes, etc.). The network structure is COV(1, 64, 5, 2, 2)-LR-COV(64, 128, 5, 2, 2)-IN-LR-COV (128, 256, 5, 2, 2)-IN-LR-COV (256, 512, 4, 1, 0)-Flatten-

LN-LR-Linear(512, 10). For every method, we set the number of training epochs to 60, maximum noise level to 0.3 for IMA, MMA and TRADES, and batch-size to 128. Adam optimizer was used with default parameters. Besides adversarial training, we did not use any other data-augmentation (e.g. crop, etc.). The results on the test set are shown in Fig. 7 and reported in Table 2 and Table 3 in Appendix B.

We also used SVHN dataset which contains 32×32 color images of $0 \sim 9$ digits. The network structure is COV(3, 32, 3, 1, 1)-LR-COV(32, 32, 3, 2, 1)-IN-LR-COV(32, 64, 3, 1, 1)-IN-LR-Conv2d(64, 64, 3, 2, 1)-IN-LR-COV(64, 128, 3, 1, 1)-IN-LR-Conv2d(128, 128, 3, 2, 1)-IN-LR-COV(128, 256, 4, 1, 0)-Flatten-LN-LR-Linear(256, 10). Besides adversarial training, we did not use any other data-augmentation. Training parameters (e.g. epochs) are the same as those for Fashion-MNIST. The results on the test set are shown in Fig. 8 and reported in Table 4 and Table 5 in Appendix B.

Form the results shown in Fig. 7 and Fig. 8 on the two datasets, it can be seen that "ce" had the worst performance on noisy samples, "adv $\varepsilon$" is very sensitive to the maximum noise level $\varepsilon$ for training, and TRADES did not perform well on L-inf norm-based tests. Compared to "ce", accuracy of every adversarial training method on clean data is slightly lower, which shows the sign of trade-off between robustness and accuracy on clean data.

Recall that IMA significantly outperformed the other methods on the Moons dataset, which demonstrates its strong potential. On Fashion-MNIST and SVHN datasets, IMA also significantly outperformed TRADES and "adv $\varepsilon$". However, IMA is only slightly better than MMA on Fashion-MNIST and on-par with MMA on SVHN. The non-ideal performance of IMA is hightly likely caused by data scarcity in high dimensional space, so that the ideal equilibrium cannot be reached. To further improve its performance, generative adversarial training could be used to generate more clean samples to fill the "holes" of the space, which will be our future work. Nevertheless, our method IMA is at least on-par with MMA method which is state-of-the-art.

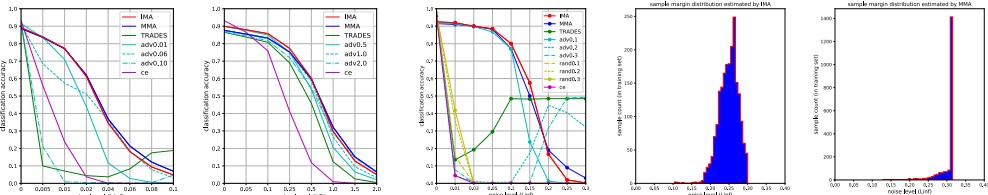

Figure 8: Results on SVHN (L-inf and L2 norm in 100-PGD).

Figure 9: Results on COVID-19 CT: accuracies (left), IMA margin estimations (middle), and MMA margin estimations(right). MMA significantly overestimated the margins.

## 3.3 APPLICATION OF COVID-19 DETECTION FROM CT IMAGES

We used a public COVID-19 CT image dataset (Soares et al., 2020), which is collected from patients in hospitals from Sao Paulo, Brazil. It contains 1252 CT scans (2D images) that are positive for COVID-19 infection and 1230 CT scans (2D images) for patients non-infected by COVID-19, 2482 CT scans in total. From infected cases, we randomly selected 200 samples for testing, 30 for validation, and 1022 for training. From the uninfected cases, we randomly selected 200 for testing, 30 for validation and 1000 for training. Each image is resized to $224 \times 224$. Since there is no pixel segmentation mask in the dataset, we can only perform classification for COVID-19 detection on the image level.

We modified the output layer of the Resnet-18 model (He et al., 2016) for the binary classification task: uninfected (label 0) vs infected (label 1), and we also replaced batch normalization with instance normalization because it is known that batch normalization is not stable for small batch-size (Wu & He, 2018). As shown in the previous studies (Shi et al., 2020), infected regions in the images have a special pattern called ground-glass opacity.

For every method, we set the number of training epochs to 100, and maximum noise level to 0.3 for IMA, MMA and TRADES, and batch-size to 32. Adam optimizer was used with default parameters. For "ce", weight decay of 0.01 is applied with Adam (i.e., AdamW with default parameters). To measure robustness on the test set, adversarial samples on different noise levels (L-inf norm) are generated by the 100-PGD. We choose L-inf over L2 norm because L-inf norm can be used to control the maximum amplitude of noises on individual pixels, which resembles random white noises from a uniform distribution. We also tested the effect of adding uniform white noises to the images, and the trained model is denoted by "rand $\varepsilon$" where $\varepsilon$ is the maximum noise level/amplitude. "rand $\varepsilon$" can be considered a weak form of adversarial training, analog to "adv $\varepsilon$". The results on the test set are shown in Fig.9 and reported in Table 6 in the appendix. Examples of clean and noisy images are shown in Fig. 10 (see in Appendix C).

The results show that IMA has the strongest overall performance, and MMA is slightly better than IMA when noise level is larger than 0.2. As shown in Fig. 10, when noise level is larger than 0.1, the noisy images look significantly different from the clean images, and those noisy images cannot be from a well-functioning CT machine (Hamper et al., 2020) and may even look like images of infection. Thus, a model robust to noises less than the level of 0.2 is good enough for this application. Also, from Fig. 9 (right), we can see that most of the estimated margins from MMA were larger than 0.3, which means most of the samples should have been free of influence of noises of magnitude less than 0.3. However, if this were true, the accuracy of MMA in Fig. 9 (left) at noise of 0.3 should not have been significantly different from that of "adv0.3". The results show that MMA significantly overestimated the margins of samples, because that MMA does not consider the equilibrium (Fig.5) between classes.

From Section 1 and 2, it can be seen that the vanilla adversarial training ("adv $\varepsilon$") is quite good when the maximum noise level $\varepsilon$ for training is chosen "properly". The question arises: what is the proper $\varepsilon$ of "adv $\varepsilon$"? A straightforward way to find a good $\epsilon$ would be running grid-research and evaluating the performance on the validation set. However, the grid-research is impractical because adversarial training is computationally expensive and time-consuming, compared to training with cross-entropy loss on clean data. It turns out that the sample margin distribution ($\mathcal{E}$ in Algorithm 1) estimated by our method IMA indicates a good $\varepsilon = 0.1$ (Fig. 9). For this application, "adv0.1" is as competitive as other advanced methods and has a less computational cost. Although adversarial training may reduce accuracy on clean data, for this application, one can tweak the classification threshold to increase the sensitivity to infection, which is often desired.

The results also show that adding random noises to the input can improve classification accuracy on clean samples (from 0.9575 of "ce" to 0.9775 of "rand0.1"). However, random noise-based data augmentation ("rand0.1", "rand0.2" and "rand0.3") has little effect on adversarial robustness for this application, which is easy to explain: not every noise (vector) is in the direction that can drive the input across the true decision boundary.

# 4 CONCLUSION

In this study, we proposed a novel Increasing-Margin Adversarial (IMA) training method to improve the robustness of DNNs for classification tasks and applied it for the application of COVID-19 detection from CT images. Our method aims to increase the margins of the training samples to improve DNN robustness against adversarial attacks. The experiment results show that our method significantly improved DNN classification accuracy on noisy data while keeping a relatively high accuracy on clean data, and its performance is at least on-par with the state-of-the-art MMA method. We also show that the vanilla adversarial training with a properly chosen $\varepsilon$ is as competitive as other advanced methods for the COVID-19 detection application, and our method IMA can provide a good $\varepsilon$ for it from the sample margin distribution. We hope our approach may facilitate the development of robust DNNs, especially for COVID-19 diagnosis using CT images.

Note: (1) we will release the code when the paper is accepted. (2) all figures are in high resolution, please zoom in. (3) please read the appendices for more results and discussions.

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

## A    APPENDIX

It seems that adversarial noises are created by algorithms (e.g. PGD) and therefore it is only a security issue caused by hackers. In fact, random imaging noises could also be "adversarial" noises leading to wrong classifications. For the COVID-19 application, we did an additional test and found out that $2.75\%$ of the noisy samples with uniform white noises on the level of 0.05, can cause the model "ce" to make wrong classifications. $2.75\%$ is not a negligible number for this application. We note that CT imaging noises can be better described by Poisson distribution (Wang et al., 2008). However, without the hardware parameters of the CT machine, it is impossible to simulate Poisson noises. Nevertheless, adversarial robustness should be the built-in property of a model for this application, and all of the DNN models in the previous COVID-19 studies (Shi et al., 2020) should be checked and enhanced for adversarial robustness before deploying those models in clinics and hospitals.

## B    APPENDIX

Table 1: Results on Moons dataset (L-inf norm-defined noise level)

| noise | 0.0 | 0.05 | 0.1 | 0.15 | 0.2 | 0.25 | 0.3 |
|---|---|---|---|---|---|---|---|
| IMA | 1.0 | 0.999 | 0.9975 | 0.9915 | 0.9755 | 0.927 | 0.823 |
| MMA | 1.0 | 0.9985 | 0.993 | 0.957 | 0.882 | 0.772 | 0.597 |
| TRADES | 0.996 | 0.9805 | 0.9575 | 0.936 | 0.906 | 0.875 | 0.8255 |
| adv | 0.9975 | 0.985 | 0.957 | 0.9365 | 0.9095 | 0.883 | 0.822 |
| ce | 1.0 | 1.0 | 0.9975 | 0.9855 | 0.935 | 0.7995 | 0.606 |

Table 2: Results on Fashion-MNIST dataset (L-inf norm-defined noise level)

| noise | 0.0 | 0.05 | 0.1 | 0.15 | 0.2 | 0.25 | 0.3 |
|---|---|---|---|---|---|---|---|
| IMA | 0.89 | 0.789 | 0.6688 | 0.5457 | 0.4121 | 0.2715 | 0.1521 |
| MMA | 0.8955 | 0.746 | 0.6166 | 0.5051 | 0.3902 | 0.2644 | 0.1425 |
| TRADES | 0.9116 | 0.4759 | 0.4151 | 0.3722 | 0.3298 | 0.2453 | 0.0949 |
| adv0.1 | 0.9112 | 0.7739 | 0.6613 | 0.4743 | 0.2376 | 0.0602 | 0.0046 |
| adv0.2 | 0.9101 | 0.5879 | 0.5218 | 0.4885 | 0.4164 | 0.2199 | 0.0313 |
| adv0.3 | 0.9132 | 0.3637 | 0.2506 | 0.1992 | 0.1671 | 0.1364 | 0.0854 |
| ce | 0.9117 | 0.0236 | 0.0 | 0.0 | 0.0 | 0.0 | 0.0 |

Table 3: Results on Fashion-MNIST dataset (L2 norm-defined noise level)

| noise | 0.0 | 0.5 | 1.0 | 2.0 | 3.0 | 4.0 | 5.0 |
|---|---|---|---|---|---|---|---|
| IMA | 0.8881 | 0.8069 | 0.6914 | 0.4711 | 0.3046 | 0.1666 | 0.0567 |
| MMA | 0.8926 | 0.807 | 0.6927 | 0.4617 | 0.2651 | 0.1361 | 0.0526 |
| DDN | 0.8666 | 0.789 | 0.6989 | 0.5066 | 0.3409 | 0.1965 | 0.0781 |
| TRADES | 0.9176 | 0.6103 | 0.4977 | 0.3425 | 0.1857 | 0.039 | 0.0008 |
| adv1 | 0.9086 | 0.8096 | 0.6549 | 0.2726 | 0.0513 | 0.0015 | 0.0 |
| adv3 | 0.9172 | 0.641 | 0.5859 | 0.4754 | 0.3325 | 0.1558 | 0.0229 |
| adv5 | 0.9148 | 0.4851 | 0.4161 | 0.3011 | 0.2121 | 0.1419 | 0.075 |
| ce | 0.9117 | 0.2512 | 0.0033 | 0.0 | 0.0 | 0.0 | 0.0 |

Table 4: Results on SVHN dataset (L-inf norm-defined noise level)

| noise | 0.0 | 0.005 | 0.01 | 0.02 | 0.04 | 0.06 | 0.08 | 0.1 |
|---|---|---|---|---|---|---|---|---|
| IMA | 0.8919 | 0.8387 | 0.7731 | 0.6123 | 0.3461 | 0.1818 | 0.0938 | 0.0481 |
| MMA | 0.887 | 0.8357 | 0.771 | 0.6204 | 0.3693 | 0.2139 | 0.1216 | 0.0688 |
| TRADES | 0.9243 | 0.0995 | 0.0705 | 0.044 | 0.0371 | 0.0827 | 0.1748 | 0.1885 |
| adv0.01 | 0.9152 | 0.8352 | 0.7095 | 0.4401 | 0.1174 | 0.0281 | 0.0062 | 0.0017 |
| adv0.06 | 0.9105 | 0.6851 | 0.5737 | 0.5115 | 0.3599 | 0.1852 | 0.0784 | 0.0318 |
| adv0.10 | 0.9209 | 0.2112 | 0.0086 | 0.0052 | 0.0048 | 0.0022 | 0.004 | 0.0472 |
| ce | 0.932 | 0.56 | 0.2384 | 0.0368 | 0.001 | 0.0 | 0.0 | 0.0 |

Table 5: Results on SVHN dataset (L2 norm-defined noise level)

| noise | 0.0 | 0.05 | 0.1 | 0.25 | 0.5 | 1.0 | 1.5 | 2.0 |
|---|---|---|---|---|---|---|---|---|
| IMA | 0.8995 | 0.8797 | 0.8576 | 0.7731 | 0.6012 | 0.2982 | 0.128 | 0.05 |
| MMA | 0.8765 | 0.8548 | 0.8322 | 0.7518 | 0.5974 | 0.3215 | 0.1519 | 0.0654 |
| DDN | 0.8649 | 0.8432 | 0.8184 | 0.7396 | 0.5852 | 0.3183 | 0.153 | 0.0669 |
| TRADES | 0.8652 | 0.8377 | 0.8076 | 0.6903 | 0.4581 | 0.1238 | 0.0256 | 0.0047 |
| adv0.5 | 0.8986 | 0.876 | 0.849 | 0.7508 | 0.539 | 0.2062 | 0.0654 | 0.0208 |
| adv1.0 | 0.873 | 0.8513 | 0.8284 | 0.7504 | 0.5857 | 0.2654 | 0.0955 | 0.0326 |
| adv2.0 | 0.8663 | 0.8436 | 0.8175 | 0.7195 | 0.5458 | 0.2883 | 0.1454 | 0.0631 |
| ce | 0.932 | 0.8656 | 0.7588 | 0.4125 | 0.1185 | 0.0092 | 0.0006 | 0.0 |

Table 6: Results on COVID-19 CT image dataset (L-inf norm-defined noise level)

| noise | 0.0 | 0.01 | 0.03 | 0.05 | 0.1 | 0.15 | 0.2 | 0.25 | 0.3 |
|---|---|---|---|---|---|---|---|---|---|
| IMA | 0.925 | 0.92 | 0.9 | 0.885 | 0.8 | 0.575 | 0.1675 | 0.02 | 0.0025 |
| MMA | 0.915 | 0.91 | 0.9 | 0.8825 | 0.77 | 0.5 | 0.19 | 0.09 | 0.03 |
| TRADES | 0.975 | 0.135 | 0.1925 | 0.295 | 0.485 | 0.4825 | 0.485 | 0.485 | 0.485 |
| adv0.1 | 0.9175 | 0.915 | 0.8975 | 0.8675 | 0.77 | 0.2375 | 0.0125 | 0.0 | 0.0 |
| adv0.15 | 0.9275 | 0.9225 | 0.91 | 0.8975 | 0.85 | 0.74 | 0.355 | 0.035 | 0.0025 |
| adv0.2 | 0.975 | 0.04 | 0.01 | 0.005 | 0.0075 | 0.1725 | 0.4475 | 0.405 | 0.3225 |
| adv0.3 | 0.975 | 0.0775 | 0.0025 | 0.0025 | 0.0 | 0.0025 | 0.3175 | 0.49 | 0.4925 |
| rand0.1 | 0.9775 | 0.155 | 0.0 | 0.0 | 0.0 | 0.0 | 0.0 | 0.0 | 0.0 |
| rand0.2 | 0.965 | 0.3575 | 0.0 | 0.0 | 0.0 | 0.0 | 0.0 | 0.0 | 0.0 |
| rand0.3 | 0.9725 | 0.4175 | 0.0 | 0.0 | 0.0 | 0.0 | 0.0 | 0.0 | 0.0 |
| ce | 0.9575 | 0.045 | 0.0 | 0.0 | 0.0 | 0.0 | 0.0 | 0.0 | 0.0 |

## C  APPENDIX

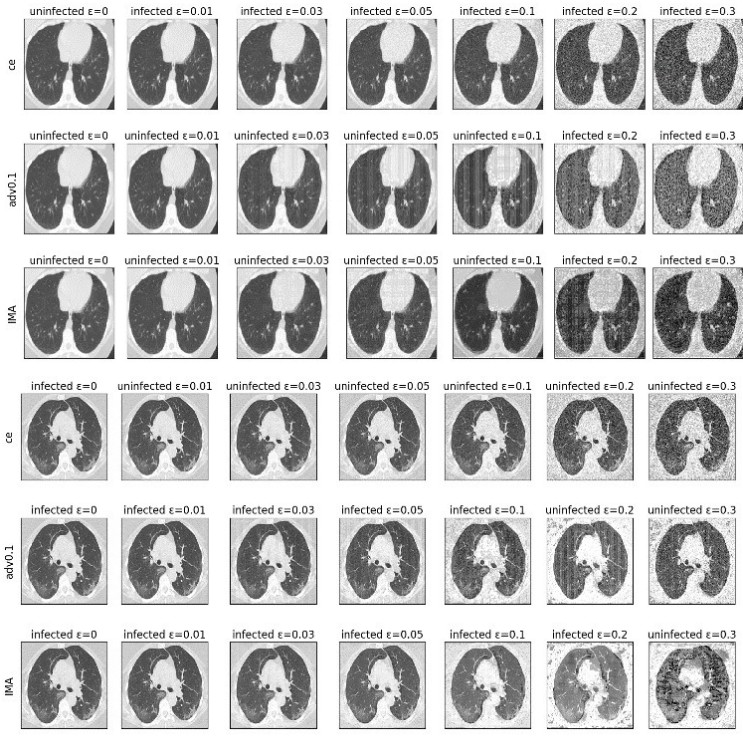

Figure 10: Each row shows a clean image and noisy images associated with a training method. The title of each image shows the predicted class label and the noise level. The clean images are correctly classified.

## D  APPENDIX

### D.1  EVALUATION ON THE MNIST DATASET

We evaluated IMA on the MNIST dataset. The maximum noise level (i.e., $\varepsilon_{max}$) matches the corresponding parameter in MMA ($4.0$ for L2-norm defined noise level and $0.45$ for Linf-defined noise level); Adamax optimizer was used with default parameters, and other settings are the same

as those for Fashion-MNIST. The CNN model is the same as that in the MMA paper. We run the IMA method twice with two different random seeds and name the resulting models as IMA_s0 and IMA_s1. The scores of MMA are directly copied from the MMA paper (Ding et al., 2019a). The settings of PGD attack are the same as those in the MMA paper. The results are reported in Table 7 and Table 8, which show that the performance of the two methods is similar.

Table 7: Results on MNIST (L2 norm-defined noise level, white-box PGD attack)

| noise | 0 | 1 | 2 | 3 | 4 |
|---|---|---|---|---|---|
| IMA_s0 | 0.9841 | 0.9387 | 0.7426 | 0.3904 | 0.0843 |
| IMA_s1 | 0.9848 | 0.9372 | 0.7379 | 0.3889 | 0.0750 |
| MMA-4.0-sd0 | 0.9871 | 0.9393 | 0.7406 | 0.4002 | 0.0581 |
| MMA-4.0-sd1 | 0.9881 | 0.9398 | 0.7381 | 0.3876 | 0.0575 |

Table 8: Results on MNIST (Linf norm-defined noise level, white-box PGD attack)

| noise | 0 | 0.1 | 0.2 | 0.3 | 0.4 |
|---|---|---|---|---|---|
| IMA_s0 | 0.9901 | 0.9764 | 0.9506 | 0.9022 | 0.7856 |
| IMA_s1 | 0.9904 | 0.9762 | 0.9508 | 0.8994 | 0.7692 |
| MMA-0.45-sd0 | 0.9895 | 0.9789 | 0.9626 | 0.9357 | 0.8816 |
| MMA-0.45-sd1 | 0.9890 | 0.9783 | 0.9618 | 0.9334 | 0.8791 |

To show that IMA does not do gradient obfuscation, we applied a black-box attack to the trained models using the SPSA method (Uesato et al., 2018). For the SPSA attack, the number of SPSA samples is 2018, the maximum number of iterations is 100, delta is 0.01 and learning rate is 0.01. Since SPSA is extremely time-consuming, we only applied it for the first 100 samples in the test set, which is the same approach used in the MMA paper. We also applied 100-PGD attack for the first 100 samples in the test set. The results are reported in Table 9 which shows that SPSA attack is not stronger than PGD attack, and therefore there is no gradient obfuscation.

Table 9: Results on MNIST subset (Linf norm-defined noise level, SPSA and PGD attacks)

| noise | 0 | 0.1 | 0.2 | 0.3 | 0.4 |
|---|---|---|---|---|---|
| SPSA on IMA_s0 | 1.00 | 1.00 | 0.99 | 0.96 | 0.96 |
| PGD on IMA_s0 | 1.00 | 1.00 | 0.98 | 0.90 | 0.82 |
| SPSA on IMA_s1 | 1.00 | 1.00 | 0.99 | 0.96 | 0.94 |
| PGD on IMA_s1 | 1.00 | 1.00 | 0.98 | 0.92 | 0.77 |

## D.2 EVALUATION ON THE CIFAR10 DATASET

We encountered a problem when we were trying to evaluate IMA and MMA on CIFAR10.

MMA(Ding et al., 2019a) was evaluated on CIFAR10 with standard data augmentation (e.g. crop). Without standard data augmentation, it is impossible to achieve a classification accuracy $> 90\%$ on clean data.

However, standard data augmentation imposes two challenges:

(1) standard data augmentation produces a random set of augmented images, and the randomness may affect robustness

(2) both IMA and MMA need to record the margin of each sample. Since the augmented images are generated online, then it is impossible to assign an ID to each augmented image. The way of MMA handling this issue is not ideal: it assigns the same ID to image $x$ and its augmented version $x\_aug$. In fact, $x$ and $x\_aug$ have different margins and thus need different IDs in the margin data array.

The solution could be to generate a large number of images from standard data augmentation and save those images to hard-drive before training. This solution needs a huge amount of storage and computing power (the margin of every sample needs to be expanded), which our humble research lab currently does not have.

Per the request of the reviewer, we evaluated the IMA method by assuming $x$ and $x\_aug$ have the same margin. $\beta$ is set to 0.5. The maximum noise level (i.e., $\varepsilon_{max}$) matches the corresponding parameter in MMA (3.0 for L2-norm-defined noise level and $32/255$ for Linf-defined noise level); SGD optimizer was used with momentum 0.9 and initial learning rate 0.1. The learning rate is reduced by half after every 25 epochs, and the total number of training epochs is 200. Since MMA used 10-PGD for adversarial training, IMA used 10 PGD-iterations in algorithm 3. The CNN model is the same as that in the MMA paper. We run the IMA method twice with two different random seeds and name the resulting models as IMA_s0 and IMA_s1. The scores of MMA are directly copied from the MMA paper. The settings of PGD attack are the same as those in the MMA paper. The results are reported in Table 10 and Table 11, which show that the performance of the two methods are similar. In general, IMA has higher accuracy on clean data, while MMA has higher accuracy on noisy data.

Table 10: Results on CIFAR10 (L2 norm-defined noise level, white-box PGD attack)

| noise | 0 | 0.5 | 1.0 | 1.5 | 2.0 | 2.5 |
|---|---|---|---|---|---|---|
| IMA_s0 | 0.8638 | 0.6584 | 0.4185 | 0.2283 | 0.1042 | 0.0375 |
| IMA_s1 | 0.8610 | 0.6549 | 0.4120 | 0.2204 | 0.0988 | 0.0389 |
| MMA-3.0-sd0 | 0.8211 | 0.6425 | 0.4761 | 0.3348 | 0.2207 | 0.1250 |
| MMA-3.0-sd1 | 0.8179 | 0.6282 | 0.4733 | 0.3379 | 0.2236 | 0.1340 |

Table 11: Results on CIFAR10 (Linf norm-defined noise level, white-box PGD attack)

| noise | 0 | 4/255 | 8/255 | 12/255 | 16/255 | 20/255 | 24/255 | 28/255 | 32/255 |
|---|---|---|---|---|---|---|---|---|---|
| IMA_s0 | 0.8678 | 0.6540 | 0.4123 | 0.2378 | 0.1259 | 0.0622 | 0.0313 | 0.0140 | 0.0073 |
| IMA_s1 | 0.8675 | 0.6489 | 0.4111 | 0.2315 | 0.1191 | 0.0628 | 0.0265 | 0.0129 | 0.0064 |
| MMA-32-sd0 | 0.8436 | 0.6525 | 0.5020 | 0.3878 | 0.3001 | 0.2257 | 0.1666 | 0.1230 | 0.0907 |
| MMA-32-sd1 | 0.8476 | 0.6466 | 0.4823 | 0.3565 | 0.2574 | 0.1786 | 0.1186 | 0.0779 | 0.0488 |

To show that IMA does not do gradient obfuscation, we applied the SPSA black-box attack to the trained models, using the same settings. The results are reported in Table 12 which shows that SPSA attack is not stronger than PGD attack, and therefore there is no gradient obfuscation.

Table 12: Results on CIFAR10 subset (Linf norm-defined noise level, SPSA and PGD attacks)

| noise | 0 | 4/255 | 8/255 | 12/255 | 16/255 | 20/255 | 24/255 | 28/255 | 32/255 |
|---|---|---|---|---|---|---|---|---|---|
| SPSA on IMA_s0 | 0.86 | 0.61 | 0.34 | 0.27 | 0.19 | 0.18 | 0.09 | 0.07 | 0.05 |
| PGD on IMA_s0 | 0.86 | 0.61 | 0.30 | 0.21 | 0.10 | 0.03 | 0.02 | 0.02 | 0.02 |
| SPSA on IMA_s1 | 0.87 | 0.63 | 0.38 | 0.26 | 0.19 | 0.12 | 0.07 | 0.07 | 0.06 |
| PGD on IMA_s1 | 0.87 | 0.61 | 0.34 | 0.21 | 0.08 | 0.06 | 0.02 | 0.02 | 0.02 |

We note that the user could tweak the parameters of IMA on the validation set to further improve the performance. In the next two appendices, Appendix E and Appendix F, we show how each of the two parameters $\beta$ and $\varepsilon_{max}$ can be used to make a trade-off between robustness and accuracy on clean data.

## E   APPENDIX

We evaluated the effect of $\beta$ in the IMA method on the Fashion-MNIST dataset, and the other settings are the same as those in Section 3.2 (e.g. white-box 100-PGD attack on the test set). The results are reported in the Table 13 and Table 14.

Table 13: Results on Fashion-MNIST (L2 norm-defined noise level)

| noise | 0 | 0.5 | 1 | 2 | 3 | 4 | 5 |
|---|---|---|---|---|---|---|---|
| $\beta$ =0.1 | 0.8963 | 0.7981 | 0.6678 | 0.4203 | 0.2597 | 0.1288 | 0.0335 |
| $\beta$ =0.3 | 0.8957 | 0.8001 | 0.6804 | 0.4469 | 0.2838 | 0.1494 | 0.0470 |
| $\beta$ =0.5 | 0.8881 | 0.8069 | 0.6914 | 0.4711 | 0.3046 | 0.1666 | 0.0567 |
| $\beta$ =0.7 | 0.8800 | 0.8075 | 0.7055 | 0.4861 | 0.3195 | 0.1829 | 0.0711 |
| $\beta$ =0.9 | 0.8691 | 0.8048 | 0.7128 | 0.5001 | 0.3340 | 0.2006 | 0.0852 |

Table 14: Results on Fashion-MNIST (Linf norm-defined noise level)

| noise | 0 | 0.05 | 0.1 | 0.15 | 0.2 | 0.25 | 0.3 |
|---|---|---|---|---|---|---|---|
| $\beta$ =0.1 | 0.8998 | 0.7638 | 0.6122 | 0.4787 | 0.3266 | 0.1919 | 0.0738 |
| $\beta$ =0.3 | 0.8945 | 0.7742 | 0.6384 | 0.5093 | 0.3596 | 0.2240 | 0.1054 |
| $\beta$ =0.5 | 0.8900 | 0.7890 | 0.6688 | 0.5457 | 0.4121 | 0.2715 | 0.1521 |
| $\beta$ =0.7 | 0.8826 | 0.7955 | 0.6864 | 0.5720 | 0.4346 | 0.2951 | 0.1745 |
| $\beta$ =0.9 | 0.8709 | 0.8029 | 0.7107 | 0.6003 | 0.4682 | 0.3265 | 0.1992 |

It can be clearly seen that smaller $\beta$ leads to higher accuracy on clean data (noise level = 0) and larger $\beta$ leads to higher accuracy on noisy data.

**The trade-off between robustness and accuracy is highly nonlinear**, as shown in the Table 15: a small decrease in accuracy on clean data can result in a large increase in accuracy on noisy data.

Table 15: Accuracy differences caused by different values of $\beta$

| noise | 0 | 0.05 | 0.1 | 0.15 | 0.2 | 0.25 | 0.3 |
|---|---|---|---|---|---|---|---|
| $\beta$ =0.1 | 0.8998 | 0.7638 | 0.6122 | 0.4787 | 0.3266 | 0.1919 | 0.0738 |
| $\beta$ =0.5 | 0.8900 | 0.7890 | 0.6688 | 0.5457 | 0.4121 | 0.2715 | 0.1521 |
| difference | 0.0098 | 0.0252 | 0.0566 | 0.0670 | 0.0855 | 0.0796 | 0.0783 |

**The nonlinear trade-off between robustness and accuracy makes it difficult to directly compare two methods, as different methods make different trade-off between robustness and accuracy.** The average accuracy is clearly not a good measure of performance on robustness.

Adjusting the parameter $\beta$ of IMA is not a computationally efficient approach to make a trade-off between robustness and accuracy, because for every possible value of $\beta$ in the range of 0 to 1, the user has to train a model from scratch through many epochs. In Appendix F, we show that the user of IMA can make such a trade-off much more efficiently by "adjusting" the parameter $\varepsilon_{max}$ of IMA.

## F   APPENDIX

In this appendix, we show how the allowed maximum margin (i.e., $\varepsilon_{max}$ in Algorithm 2) affects the performance of IMA, and how to choose its value.

Noisy samples with adversarial noises can be correctly-classified by humans but may be incorrectly-classified by neural networks, which is basically the definition of adversarial noises. By following

this definition, we choose the allowed maximum margin by looking at the noisy samples: if the noise magnitude is too large such that we barely can recognize the objects in the images, then this magnitude is chosen as the allowed maximum margin. This is how we choose the allowed maximum margin in the experiments on the datasets, except for MNIST and CIFAR, for which the allowed maximum margin matches the corresponding parameter of the MMA method.

Next, we provide a strategy for the user of IMA to refine the choice of $\varepsilon_{max}$ by making a trade-off between robustness and accuracy. As shown in Fig. 11, the trade-off between clean-accuracy (i.e., accuracy on clean data) and robustness (i.e., accuracy on noisy data) can be observed during training by using our IMA method.

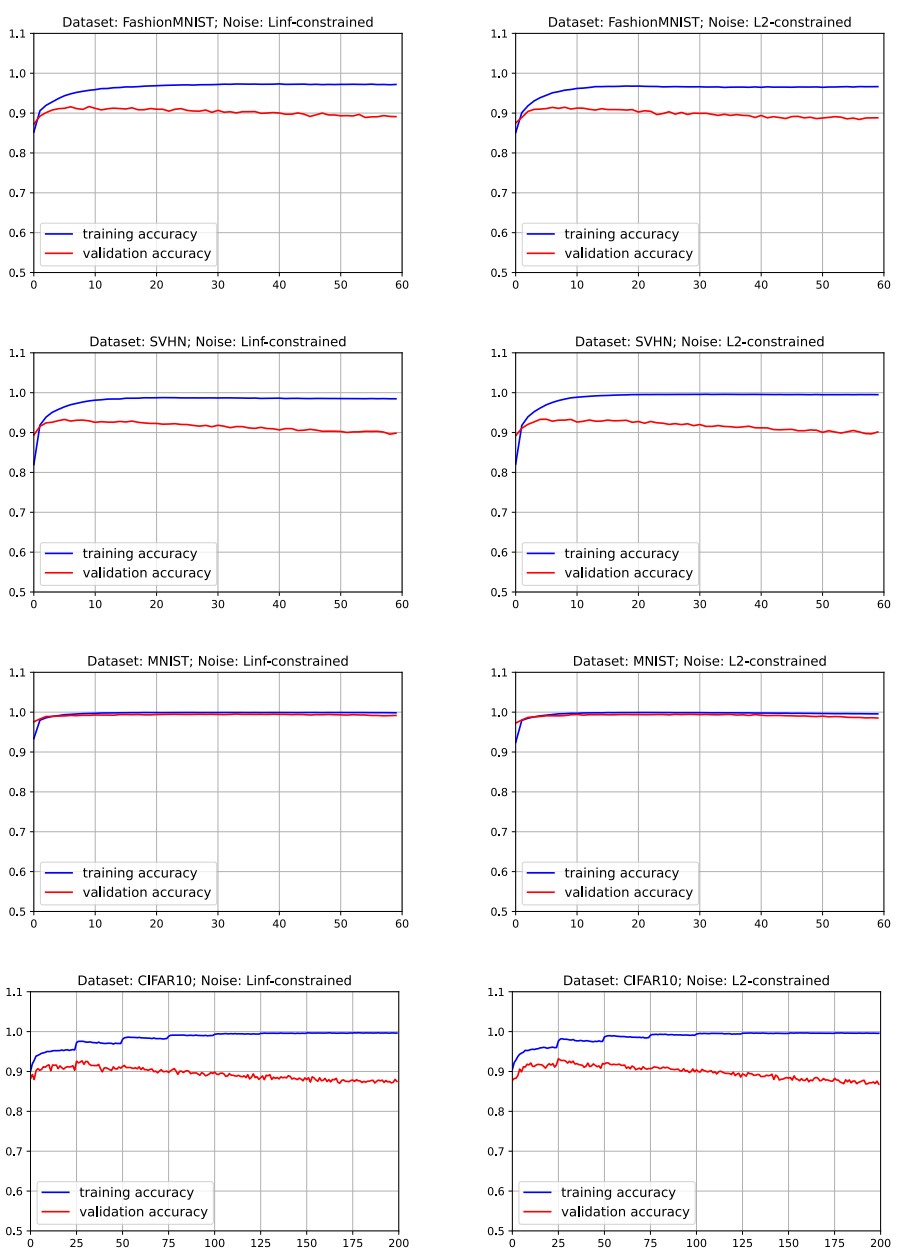

Figure 11: The training and validation curves (accuracy vs epoch) on different datasets obtained by using our IMA method. The accuracy scores are measured on clean data.

During IMA training, the (estimated) margin of a sample $x1$ in class-1 is initialized to be close to 0 and it keeps increasing as if the radius of a ball increases during the training process. The sample $x1$ is at the center of the ball, and the radius of the ball is the current margin of the sample. When the ball of $x1$ collides with the ball of another sample $x2$ from a different class-2, then a local decision boundary is formed, and the two balls stop expanding.

If there are enough training data samples, a sample $x$ in some class will eventually meet with its counterpart in another class somewhere in the middle of the two classes, which forms a robust decision boundary. In practice, the amount of training data is never enough to cover the input space, and therefore the margin of a sample could be overestimated because of missing the counterparts that are needed to stop the expansion of the margin. If the margins of many samples are overestimated, then the balls of these samples may penetrate the optimal decision boundary and cause lower classification accuracy on the validation set of clean samples, **which explains the existence of the trade-off between robustness and clean-accuracy from the perspective of sample margins.**

The above analysis is confirmed by the trend of the training and validation accuracy curves in the Fig. 11: after some epochs, the training accuracy curve becomes stable, and the validation accuracy curve starts to decrease, which indicates margin overestimation occurs. **For our IMA method, the cause of margin overestimation is the lack of data in high dimensional space.** In the 2D Moons dataset, there are enough training samples, and therefore, the decision boundary of our IMA method is almost in the "middle" between the two classes, i.e., no margin overestimation.

**Using the validation accuracy curve, the user of IMA can choose the allowed maximum margin** such that validation accuracy is above a pre-defined threshold that is set by the user to meet some application requirements. Next, we demonstrate this approach on the Fashion-MNIST dataset. We note that during IMA training, the margins of the samples are gradually increasing. Immediately after $M$ epochs, the maximum of the training sample margins is $M \times \Delta\varepsilon$ where $\Delta\varepsilon$ is the margin expansion step size in Algorithm 2. Thus, the maximum margin is $\varepsilon_{max} = M \times \Delta\varepsilon$ for the model trained for $M$ epochs. Since we do not know the threshold on validation accuracy, which a user may be interested in on the dataset, we selected six models trained by IMA after 10, 20, 30, 40, 50, and 60 epochs, and we evaluated the robustness of the six models on the test set by using the 100-PGD attack. Basically, we re-analyzed the models trained through 60 epochs in Section 3.2. The results are reported in Table 16 and Table 17, which reveal the effect of $\varepsilon_{max}$ on robustness.

Table 16: Effect of $\varepsilon_{max}$ on Fashion-MNIST test set (L2 norm-defined noise level, $\Delta\varepsilon = 5/60$). The last column (named 0 (val)) shows the accuracies on the validation set (clean data).

| noise | 0 | 0.5 | 1 | 2 | 3 | 4 | 5 | 0 (val) |
|---|---|---|---|---|---|---|---|---|
| $\varepsilon_{max} = 10\Delta\varepsilon$ | 0.9106 | 0.7760 | 0.5590 | 0.1275 | 0.0037 | 0 | 0 | 0.9102 |
| $\varepsilon_{max} = 20\Delta\varepsilon$ | 0.9058 | 0.8103 | 0.6758 | 0.3448 | 0.0831 | 0.0023 | 0 | 0.9087 |
| $\varepsilon_{max} = 30\Delta\varepsilon$ | 0.8976 | 0.8122 | 0.6975 | 0.4468 | 0.2070 | 0.0441 | 0.0019 | 0.8998 |
| $\varepsilon_{max} = 40\Delta\varepsilon$ | 0.8888 | 0.8116 | 0.7061 | 0.4805 | 0.2915 | 0.1267 | 0.0289 | 0.8890 |
| $\varepsilon_{max} = 50\Delta\varepsilon$ | 0.8887 | 0.8058 | 0.6968 | 0.4772 | 0.3153 | 0.1685 | 0.0538 | 0.8862 |
| $\varepsilon_{max} = 60\Delta\varepsilon$ | 0.8881 | 0.8069 | 0.6914 | 0.4711 | 0.3046 | 0.1666 | 0.0567 | 0.8882 |

Table 17: Effect of $\varepsilon_{max}$ on Fashion-MNIST test set (Linf norm-defined noise level, $\Delta\varepsilon = 0.3/60$). The last column (named 0 (val)) shows the accuracies on the validation set (clean data)

| noise | 0 | 0.05 | 0.1 | 0.15 | 0.2 | 0.25 | 0.3 | 0 (val) |
|---|---|---|---|---|---|---|---|---|
| $\varepsilon_{max} = 10\Delta\varepsilon$ | 0.9159 | 0.6479 | 0.2598 | 0.0402 | 0.0009 | 0 | 0 | 0.9165 |
| $\varepsilon_{max} = 20\Delta\varepsilon$ | 0.9095 | 0.7459 | 0.5165 | 0.2449 | 0.0647 | 0.0035 | 0 | 0.9115 |
| $\varepsilon_{max} = 30\Delta\varepsilon$ | 0.9015 | 0.7754 | 0.6021 | 0.4022 | 0.2050 | 0.0644 | 0.0053 | 0.9018 |
| $\varepsilon_{max} = 40\Delta\varepsilon$ | 0.8995 | 0.7895 | 0.6452 | 0.4877 | 0.3141 | 0.1588 | 0.0446 | 0.9013 |
| $\varepsilon_{max} = 50\Delta\varepsilon$ | 0.8957 | 0.7887 | 0.6564 | 0.5217 | 0.3657 | 0.2245 | 0.0989 | 0.8953 |
| $\varepsilon_{max} = 60\Delta\varepsilon$ | 0.8900 | 0.7890 | 0.6688 | 0.5457 | 0.4121 | 0.2715 | 0.1521 | 0.8910 |

From Table 16 and Table 17, it can be clearly seen that smaller $\varepsilon_{max}$ leads to higher accuracy on clean data (noise level = 0), and larger $\varepsilon_{max}$ leads to higher accuracy on noisy data. Since the maximum of the sample margins gradually increases during IMA training, the validation accuracy changes gradually (increasing and then decreasing), which makes it easy for the user of our method to choose the trained model with validation accuracy above the user-defined threshold.

We have done similar analyses on the models trained by IMA through 60 epochs on the SVHN dataset (Section 3.2). We selected six models trained by IMA after 10, 20, 30, 40, 50, and 60 epochs, and we evaluated the robustness of the six models on the test set by using the 100-PGD attack. The results are reported in Table 18 and Table 19, which reveal the effect of $\varepsilon_{max}$ on robustness. By monitoring the validation accuracy curve during training, the user of IMA can choose a trained model such that validation accuracy of the chosen model is above a threshold defined by the user to meet some application requirements.

Table 18: Effect of $\varepsilon_{max}$ on SVHN test set (L2 norm-defined noise level, $\Delta\varepsilon = 2/60$). The last column (named 0 (val)) shows the accuracies on the validation set (clean data).

| noise | 0 | 0.05 | 0.1 | 0.25 | 0.5 | 1 | 1.5 | 2 | 0 (val) |
|---|---|---|---|---|---|---|---|---|---|
| $\varepsilon_{max} = 10\Delta\varepsilon$ | 0.9382 | 0.9173 | 0.8888 | 0.7539 | 0.4602 | 0.1131 | 0.0231 | 0.0040 | 0.933 |
| $\varepsilon_{max} = 20\Delta\varepsilon$ | 0.9303 | 0.9126 | 0.8893 | 0.7918 | 0.5658 | 0.2002 | 0.0570 | 0.0147 | 0.9257 |
| $\varepsilon_{max} = 30\Delta\varepsilon$ | 0.9204 | 0.9012 | 0.8797 | 0.7944 | 0.6020 | 0.2588 | 0.0918 | 0.0290 | 0.9173 |
| $\varepsilon_{max} = 40\Delta\varepsilon$ | 0.9139 | 0.8952 | 0.8721 | 0.7885 | 0.6081 | 0.2820 | 0.1066 | 0.0371 | 0.9120 |
| $\varepsilon_{max} = 50\Delta\varepsilon$ | 0.9014 | 0.8824 | 0.8616 | 0.7769 | 0.6083 | 0.3002 | 0.1258 | 0.0484 | 0.9058 |
| $\varepsilon_{max} = 60\Delta\varepsilon$ | 0.8995 | 0.8797 | 0.8576 | 0.7731 | 0.6012 | 0.2982 | 0.1280 | 0.0500 | 0.9016 |

Table 19: Effect of $\varepsilon_{max}$ on SVHN test set (Linf norm-defined noise level, $\Delta\varepsilon = 0.1/60$). The last column (named 0 (val)) shows the accuracies on the validation set (clean data).

| noise | 0 | 0.005 | 0.01 | 0.02 | 0.04 | 0.06 | 0.08 | 0.1 | 0 (val) |
|---|---|---|---|---|---|---|---|---|---|
| $\varepsilon_{max} = 10\Delta\varepsilon$ | 0.9362 | 0.8674 | 0.7497 | 0.4637 | 0.1258 | 0.0284 | 0.0060 | 0.0012 | 0.9293 |
| $\varepsilon_{max} = 20\Delta\varepsilon$ | 0.9294 | 0.8708 | 0.7869 | 0.5756 | 0.2409 | 0.0850 | 0.0291 | 0.0091 | 0.9229 |
| $\varepsilon_{max} = 30\Delta\varepsilon$ | 0.9171 | 0.8617 | 0.7878 | 0.6095 | 0.3035 | 0.1317 | 0.0543 | 0.0208 | 0.9146 |
| $\varepsilon_{max} = 40\Delta\varepsilon$ | 0.9110 | 0.8566 | 0.7862 | 0.6137 | 0.3259 | 0.1543 | 0.0702 | 0.0320 | 0.9095 |
| $\varepsilon_{max} = 50\Delta\varepsilon$ | 0.9049 | 0.8518 | 0.7836 | 0.6180 | 0.3418 | 0.1724 | 0.0850 | 0.0417 | 0.9034 |
| $\varepsilon_{max} = 60\Delta\varepsilon$ | 0.8919 | 0.8387 | 0.7731 | 0.6123 | 0.3461 | 0.1818 | 0.0938 | 0.0481 | 0.8983 |

We also have done similar analyses on the models trained by IMA through 200 epochs on the CI-FAR10 dataset (Appendix D.2). We selected four models trained by IMA after 50, 100, 150, and 200 epochs, and we evaluated the robustness of the four models on the test set by using the 100-PGD attack. The results are reported in Table 20 and Table 21, which reveal the effect of $\varepsilon_{max}$ on robustness. By monitoring the validation accuracy curve during training, the user of IMA can choose a trained model such that validation accuracy of the chosen model is above a threshold defined by the user to meet some application requirements.

Table 20: Effect of $\varepsilon_{max}$ on CIFAR10 test set (L2 norm-defined noise level, $\Delta\varepsilon$=3/200). The last column (named 0 (val)) shows the accuracies on the validation set (clean data).

| noise | 0 | 0.5 | 1 | 1.5 | 2 | 2.5 | 0 (val) |
|---|---|---|---|---|---|---|---|
| $\varepsilon_{max} = 50\Delta\varepsilon$ | 0.9140 | 0.5616 | 0.1792 | 0.0324 | 0.0036 | 0.0004 | 0.9124 |
| $\varepsilon_{max} = 100\Delta\varepsilon$ | 0.8959 | 0.6545 | 0.3547 | 0.1461 | 0.0451 | 0.0122 | 0.8980 |
| $\varepsilon_{max} = 150\Delta\varepsilon$ | 0.8728 | 0.6644 | 0.4037 | 0.2041 | 0.0873 | 0.0333 | 0.8798 |
| $\varepsilon_{max} = 200\Delta\varepsilon$ | 0.8610 | 0.6549 | 0.4120 | 0.2204 | 0.0988 | 0.0389 | 0.8676 |

Table 21: Effect of $\varepsilon_{max}$ on CIFAR10 test set (Linf norm-defined noise level, $\Delta\varepsilon=(32/255)/200$). The last column (named 0 (val)) shows the accuracies on the validation set (clean data).

| noise | 0 | 4/255 | 8/255 | 12/255 | 16/255 | 20/255 | 24/255 | 28/255 | 32/255 | 0 (val) |
|---|---|---|---|---|---|---|---|---|---|---|
| $\varepsilon_{max} = 50\Delta\varepsilon$ | 0.9079 | 0.6056 | 0.2756 | 0.0911 | 0.0258 | 0.0073 | 0.0019 | 0.0004 | 0.0002 | 0.9090 |
| $\varepsilon_{max} = 100\Delta\varepsilon$ | 0.8932 | 0.6598 | 0.3983 | 0.2107 | 0.1050 | 0.0494 | 0.0236 | 0.0113 | 0.0059 | 0.8974 |
| $\varepsilon_{max} = 150\Delta\varepsilon$ | 0.8758 | 0.6598 | 0.4170 | 0.2369 | 0.1248 | 0.0624 | 0.0301 | 0.0146 | 0.0082 | 0.8816 |
| $\varepsilon_{max} = 200\Delta\varepsilon$ | 0.8675 | 0.6489 | 0.4111 | 0.2315 | 0.1191 | 0.0628 | 0.0265 | 0.0129 | 0.0064 | 0.8704 |

Next, we plot the curves from the COVID-19 dataset in Fig. 12. The validation accuracy decreased only slightly after 20 epochs, which means the value of the allowed maximum margin is reasonable and the risk of margin overestimation is low.

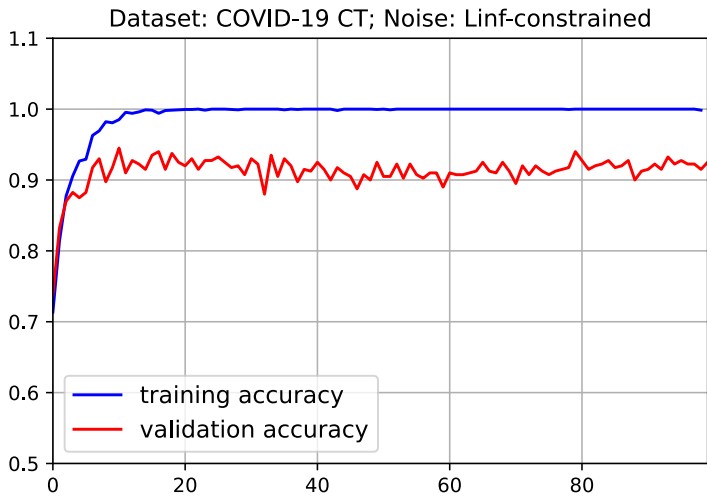

Figure 12: The training and validation curves (accuracy vs epoch) on the COVID-19 CT dataset obtained by using our IMA method. The accuracy scores are measured on clean data.

We note that **using the margin distribution estimated by IMA, we can obtain a good perturbation mangnitude $\varepsilon$ (0.1 or 0.15) for the vanilla adversarial training. When $\varepsilon$=0.15, vanilla adversarial training achieved the best performance on the COVID-19 CT dataset, compared to other methods.** Please see Fig. 14 which contains the margin distributions estimated by IMA, and Table 6 which contains accuracy scores in a large range of noise levels. In practice, we only need robustness against noises within a certain level, because significantly-noisy images can only be produced from a malfunctioning CT machine. Thus, vanilla adversarial training (denoted by adv $\varepsilon$) should be good enough (much less computation cost and time compared to advanced adversarial training methods) as long as its parameter $\varepsilon$ is appropriate. In this application, good values of $\varepsilon$ are 0.1 and 0.15, as revealed by the margin distribution estimated by IMA. For the convenience of the reader, we re-plot the results in the new Fig. 13 and Fig. 14.

For vanilla adversarial training, a straightforward way to find a good $\varepsilon$ would be running grid-research and evaluating the performance on the validation set. However, the grid-research in a large range (e.g. 0 to 0.3) is impractical because adversarial training is computationally expensive and time-consuming, compared to standard training with cross-entropy loss and clean data. The margin distribution estimated by IMA can reveal a good range of the $\varepsilon$, and then grid-research in this small range can be performed to find the optimal $\varepsilon$. After the optimal $\varepsilon$ is found for vanilla adversarial training, we could could combine other techniques to further improve robustness.

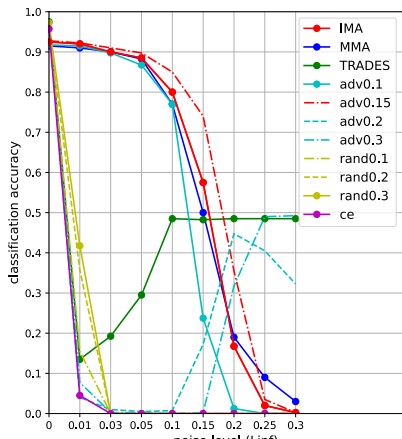
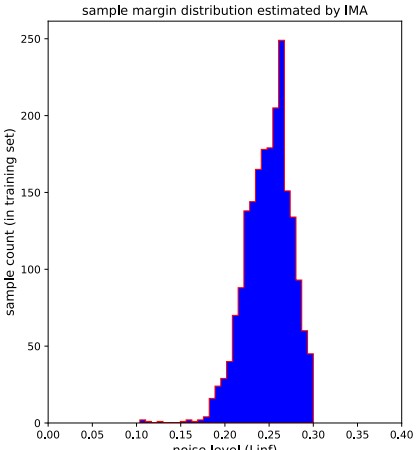

Figure 13: The performance of the methods on COVID-19 test set

Figure 14: Margin distribution estimated by IMA on training set

# G APPENDIX

## G.1 THE BASIC IDEA OF OUR IMA METHOD

If there are only two classes and the data samples are linearly-separable, then linear SVM (support vector machine) will produce a linear decision boundary in the "middle" between the two classes. The decision boundary of SVM is robust against noises: the classification output will not change if a small amount of noise $\delta$ is added to $x$ as long as the vector norm of $\delta$ is smaller than the margin of $x$. Here, the margin of $x$ is the (minimum) distance between $x$ and the decision boundary.

In general, the data samples in multiple classes are nonlinearly-separable, and the robust decision boundary should be somewhere in the middle between classes, which is the goal that IMA pursues. The result on the Moons dataset shows that IMA can indeed produce a nonlinear decision boundary in the "middle" between classes. We use 2D Moons dataset because it is impossible to directly visualize a nonlinear decision boundary in a high dimensional space. As shown on this dataset, other methods are not trying to find a decision boundary in the "middle" of the two classes.

## G.2 THE EQUILIBRIUM STATE

Our IMA method is a heuristic-based method that is not derived from any theory. We use the equilibrium state analysis to provide a theoretical explanation of the method. We have shown that an equilibrium state can be achieved when the noisy samples have the same spatial distribution on the decision boundary. Here, we will analyze what will happen if the spatial distributions of the noisy samples in different classes are not the same on the current decision boundary. We note that our IMA method will actively generate and put noisy samples on ("close to" due to numerical precision) the current decision boundary of the neural network model, and the training is a dynamic process to adjust the decision boundary of the model. Let's focus on the following two terms:

$$F_i \triangleq E_{X_n \in c_i \ and \ X_n \in B_{ij}} = -\int q_i(x) log(P_i(x)) dx \tag{9}$$

$$F_j \triangleq E_{X_n \in c_j \ and \ X_n \in B_{ij}} = -\int q_j(x) log(P_j(x)) dx \tag{10}$$

where $q_i(x)$ and $q_j(x)$ are the distributions (i.e., densities) of the noisy samples on the current decision boundary between the two classes, and $q_i(x)$ and $q_j(x)$ may not be equal to each other.

In fact, $F_i$ and $F_j$ can be interpreted as two forces that try to expand the margins of the samples in the two classes against each other. By dividing the decision boundary into small regions (i.e., linear segments), the two integrals can be evaluated in the individual regions. In a region, if $q_i(x) > q_j(x)$ (i.e., more samples in class-i) then the current state is not in equilibrium: after updating the model using these noisy samples, the noisy samples in class-i will be correctly classified and the noisy samples in class-j will be incorrectly-classified (this is a simple result of classification with imbalanced data in the region), which means the decision boundary will shift towards the samples in class-j, and therefore the margins of the corresponding samples in class-i will expand and the margins of the corresponding samples in class-j will shrink. Thus, the decision boundary may shift locally towards the samples in one of the classes. Obviously, the decision boundary will stop shifting when the local densities of noisy samples in different classes are the same along the decision boundary, i.e., $q_i(x)$ becomes equal to $q_j(x)$, which means an equilibrium state is reached. This analysis is a sketch of the dynamic process during training, and we defer a math-rigor analysis to our future work. Two questions arise from the above analysis:

(1) Will there exist a decision boundary somewhere in the "middle" between classes, along which the local densities of noisy samples in different classes are roughly the same? Clearly, the existence of such a decision boundary is determined by the dataset itself. If the samples in different classes are well separated (e.g. the datasets used in our experiments), we hypothesize that such a decision boundary exists, just like the decision boundary of IMA on the 2D Moons dataset.

(2) Assuming such a robust decision boundary exists on a dataset, will the IMA method find it? Although it cannot give a theoretical guarantee, the experimental results show that our IMA method can indeed increase the margins of the samples on different datasets: increasing margin means increasing robustness.

## H  APPENDIX

In this appendix, we discuss the trade-off between robustness and accuracy from the perspective of sample margins. Let $x$ denote a clean sample (i.e., unaltered by any adversarial attacks). Let $x_{(\delta)}$ denote the noisy sample generated by adding an adversarial noise $\delta$ to $x$, i.e., $x_{(\delta)} = x + \delta$. The vector norm of $\delta$ is $\varepsilon$. Let $y$ denote the true class label of $x$.

For vanilla adversarial training, the class label of the noisy sample $x_{(\delta)}$ is assumed to be the same as the class label $y$ of the clean sample $x$, no matter how large the noise level $\varepsilon$ is. This label assignment can be wrong if $\varepsilon$ is very large: large enough such that $x_{(\delta)}$ may "look like" a sample in a different class (not $y$), which has been shown in numerous studies. Here, "look like" refers to the judgment of a human. However, it is impractical to let a person actually look at every noisy sample to assign the right label to it.

If there are unlimited training samples that cover the input space, the samples can reveal the true class distributions $p(x|y)$. Then, the optimal decision boundary can be obtained by Bayesian classification using $p(x|y)$ and $p(y)$ of each of the classes. For example, $p(x|y)$ can be simply modeled by a Gaussian mixture model, and $p(y)$ may be assumed to be the same for every class. If we assume Bayesian classification with unlimited data is as good as the judgment of human experts, then Bayesian decision boundary is optimal and robust against noises. Let $g(x)$ denote the Bayesian classifier ($g$ means ground truth), and it outputs the true label of the input: $y = g(x)$ and $y_{(\delta)} = g(x_{(\delta)})$. Using the Bayesian-optimal decision boundary, the true margin of a clean sample $x$ can be obtained, i.e., the (minimum) distance between $x$ and the decision boundary, which is denoted by $m(x)$.

During adversarial training (vanilla or IMA), a noisy sample is generated, $x_{(\delta)} = x + \delta$ with $\varepsilon = ||\delta||$, and the class label $y$ is assigned to it. If the noise/perturbation magnitude $\varepsilon$ is larger then the true margin $m(x)$, then the assigned class label is wrong for $x_{(\delta)}$, and training the model with $x_{(\delta)}$ may cause its decision boundary to deviate from the Bayesian-optimal decision boundary even if the model $f(x)$ is initialized to be very close to $g(x)$. As a result, the classification accuracy on the test/validation set (clean data) will become lower, but the classification accuracy on the training set (clean data) may not change because the decision boundary has been pushed far away from the clean training sample $x$. **This phenomenon has been revealed by the training and validation accuracy curves of IMA (see Fig. 11 in Appendix F), which explains the well-known trade-off between robustness and accuracy.**

Thus, the success of adversarial training depends on the accurate estimation of the true margin $m(x)$: if $\mathcal{E}(x) = m(x)$, then IMA could be perfect. For the 2D Moons dataset, IMA can indeed find a nearly perfect decision boundary, significantly better than the other methods (see Fig. 6 in Section 3.1). However, for a high dimensional dataset (e.g. Fashion-MNIST), there are not enough training samples to cover the input space. As a result, perfect Bayesian classification is not achievable because of inaccurate estimation of $p(x|y)$, and IMA cannot obtain perfect estimations of the true margins because there may not exist the counterparts in other classes (not $y$) that can stop the expansion of $\varepsilon$-ball (margin estimation) of $x$ in class $y$. Margin overestimation is revealed by the gradually-decreasing trend of the validation accuracy curve (see Fig. 11 in Appendix F). However, the good news for the user of IMA is that the user can choose a good value of $\varepsilon_{max}$ such that the validation accuracy is above a pre-defined threshold (e.g. 90% for the COVID-19 application). Also, the user does not need to pre-define the value of $\varepsilon_{max}$: the allowed maximum margin will increase with the number of training epochs (see Appendix F), and the user only needs to monitor the training and validation accuracy curves. Although IMA is not perfect, it brings such a significant **benefit to the user: the ease of "adjusting" $\varepsilon_{max}$ to make a trade-off between robustness and accuracy (simply set $\beta$=0.5).**

From IMA, when an equilibrium state is reached, the distributions (i.e., local densities) of noisy samples in different classes are the same along the decision boundary (note: it does not necessarily mean the clean samples in different classes are equally spaced from the decision boundary). This is somewhat analog to Bayesian classification: at the optimal decision boundary, the distributions (densities) of samples in two classes are the same, assuming the classes have equal prior probabilities. From this perspective, the noisy samples, which are generated by IMA, serve as the surrogates of the real samples. Obviously, we cannot claim it is Bayesian classification because noisy samples may not reveal the true distributions. From this perspective, more advanced adversarial training methods may be developed such that the generated samples may reveal the true distributions (i.e., $p(x|y)$); if so, then the resulting decision boundary could be optimal and robust.

Additional Note: The validation accuracy curves of IMA in Fig. 11 in Appendix F increase and then gradually decrease. Please do not confuse this with the common concept of overfitting on clean data. Actually, when the models were trained with cross-entropy loss and clean data, there is no gradually-decreasing trend in validation accuracy curves, as shown in Fig. 15. Thus, the only explanation of the gradually-decreasing trend in Fig. 11 in Appendix F is margin overestimation.

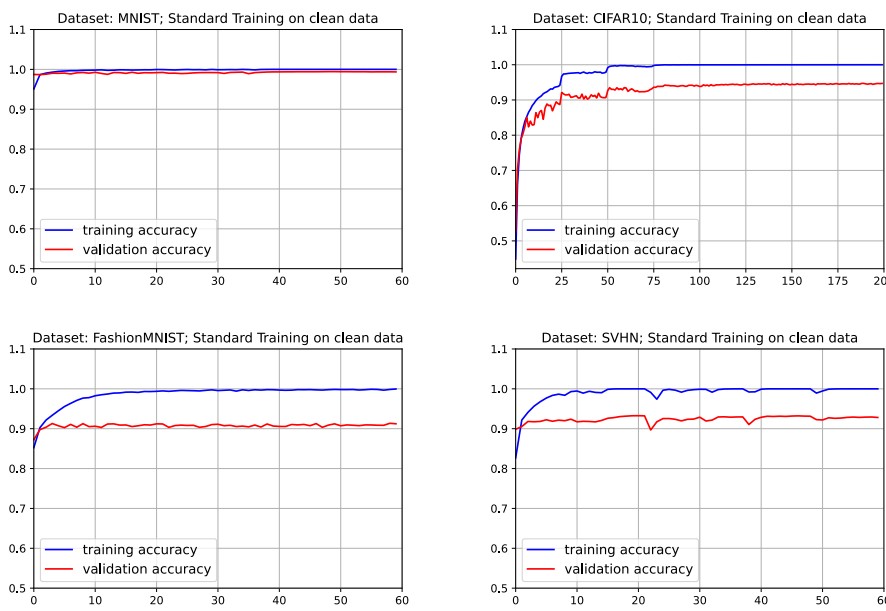

Figure 15: Training and validation accuracy curves (accuracy vs epoch) on the datasets, using standard training with cross-entropy loss and clean data. The accuracy scores are measured on clean data.

