# OpenReview forum: "Increasing-Margin Adversarial (IMA) training to Improve Adversarial Robustness of Neural Networks"
_ICLR.cc/2021/Conference — Reject_

### Official Review · AnonReviewer1 · 2020-10-23
**Minor novelty, not sufficiently clear presentation**

**Rating:** 4
**Confidence:** 1

**Review:**

The paper proposes to increase the adversarial robustness of a neural net by training the model on both clean and adversarial samples. An adaptive form of the projected gradient descent generates the adversarial samples. Therefore, the noise magnitude is estimated separately for each training sample, such that the decision boundary (suppose a classification problem) of the neural net has maximum distance to each training sample.

Strengths:
1.	Appealing idea of having adaptive noise magnitudes.
2.	Relevant experimental section (Covid19).
3.	Illustrative figures, describing the model.

Weaknesses, Suggestions, Questions:
1.	A theoretical discussion about following points will improve the contribution of the paper:
        a.	Why do large margins result in higher adversarial robustness? What happens if I change the attack type?
        b.	Benefits compared over other adversarial training methods are not clear.
        c.	A more detailed discussion about the equilibrium state is necessary, as currently provided in Sec. 2.3. This is rather an example.
2.	Experimental section:
        a.	Need to report average over multiple runs. Results are very close together and it is hard to favor one method.
        b.	Sec. 3.1: Since this is the toy-dataset, a discussion why the decision boundaries look as they do, would be interesting.
        c.	Sec. 3.3: What information is in Fig. 9 middle and right?
3.	Formatting and writing:
        a.	Detailed proofreading required.  e.g. on p. 3  “using cross-entropy loss and clean data for training”
        b.	Some variables are used but not introduced.  e.g. x_n1, x_n2  in Sec. 2.3.
        c.	Figures are too small and not properly labeled in experimental section.
        d.	References to prior work are missing as e.g. “Virtual Adversarial Training: A Regularization Method for Supervised and Semi-Supervised Learning”
        e.	Algorithms need rework, e.g. information of Alg. 1 can be written in 2,3 lines.

Though the idea of adaptive adversarial noise magnitude is in general appealing, the paper has some weaknesses: (i) theoretical contribution is relatively minor, (ii) the paper does not present the material sufficiently clearly to the reader, and (iii) experimental evaluation is not sufficiently conclusive in favor of the paper's central hypothesis.

---

> ### Author Response · Authors · 2020-11-22
> **Reply to AnonReviewer1**
>
> (1) "Why do large margins result in higher adversarial robustness? What happens if I change the attack type? "
>
> Reply: Since everyone is familiar with SVM (support vector machine), here, we use SVM to explain why larger margins result in higher adversarial robustness.
>
> If there are only two classes and the data samples are linearly-separable, then (linear) SVM will produce a linear decision boundary in the "middle" between the two classes. The decision boundary of SVM is robust against noises: the classification output will not change if a small amount of noise δ is added to x as long as the vector norm of δ is smaller than the margin of x. The decision boundary is also robust to noises from any type of adversarial attacks, as long as the vector norm of δ is smaller than the margin of x. Here, the margin of x is the (minimum) distance between x and the decision boundary.
>
> In general, the data samples in multiple classes are nonlinearly-separable, and the robust decision boundary should be somewhere in the middle between classes, which is the goal that IMA pursues.
>
> (2) "Benefits compared over other adversarial training methods are not clear."
>
> Reply: IMA outperforms vanilla adversarial training and TRADES, and it is on par with MMA in the experiments.
> The benefits of the IMA are discussed in appendices (F, G, and H) of the revised manuscript. Here is a brief summary.
>
> For algorithm designers/researchers: the IMA method and the experimental results explain the trade-off between robustness and accuracy from the perspective of sample margins (see Appendices F and G). Our work has demonstrated that "common intuition that adversarial attacks are most influential to the points close to the decision boundary" can be materialized into algorithms to improve adversarial robustness, which points out a promising direction for defense against adversarial attacks.
>
> For the user of IMA to make robust applications (e.g. COVID-19 CT): the IMA method provides a convenient and efficient way to make a trade-off between robustness and accuracy (see Appendices F and G), which is difficult to do for other methods.
>
> (3) "A more detailed discussion about the equilibrium state is necessary, as currently provided in Sec. 2.3. This is rather an example."
>
> Reply: Equations (3) to (8) show the equilibrium state when there are three classes, and it is mathematically trivial to show it is also true for more than three classes: we only need to focus on a pair of classes at a time. In Appendix G, we provide further explanation of the IMA method.
>
> (3) "Need to report average over multiple runs. Results are very close together and it is hard to favor one method."
>
> Reply: it would be great to do multiple runs and get a p-value. However, few people in the field has done this because of high computation cost (experiments on a dataset may take weeks). In Appendix D, we add additional two experiments on MNIST and CIFAR10, and each one runs twice with different random seeds, just like what was done in the MMA paper.
>
> (4) "Sec. 3.1: Since this is the toy-dataset, a discussion why the decision boundaries look as they do, would be interesting."
>
> Reply: In general, the data samples in multiple classes are nonlinearly-separable, and the robust decision boundary should be somewhere in the middle between classes, similar to SVM. And the decision boundary of IMA is indeed roughly in the middle. Other methods are not trying to get such a decision boundary.
>
> (5) "Sec. 3.3: What information is in Fig. 9 middle and right?"
>
> Reply: As explained by the captions, Fig. 9 middle shows the sample margin distribution estimated by IMA. Fig. 9 right shows the sample margin distribution estimated by MMA, which indicates significant overestimation because MMA-estimated margin distribution is significantly in contradiction with MMA accuracy scores on the noisy data.
>
> (6) “using cross-entropy loss and clean data for training”"
>
> Reply: thank you for this advice. in the revised paper, we changed it to using standard training with cross-entropy loss and clean data

---

> > ### Author Response · Authors · 2020-11-22
> > **continue...**
> >
> >
> > (7) " Some variables are used but not introduced.  e.g. x_n1, x_n2  in Sec. 2.3."
> >
> > Reply: the two variables are defined in Algorithm 3.
> >
> > (8) Figures are too small and not properly labeled in experimental section.
> >
> > Reply: due to the 8-page limit, we have to squeeze some figures together. The figures are embedded into pdf with high resolution, please zoom in on computer screen.
> >
> > (9) " References to prior work are missing as e.g. “Virtual Adversarial Training: A Regularization Method for Supervised and Semi-Supervised Learning”
> >
> > Reply: we cited this paper in the revised manuscript. Given the large number of papers published per year, it is infeasible to cite everyone. We cited the most relevant papers that we compared with, and we cited several reviewer papers in the introduction section.  Please let us know if any other paper should be cited.
> >
> > (9) "Algorithms need rework, e.g. information of Alg. 1 can be written in 2,3 lines."
> >
> > Reply: we thank the reviewer for this suggestion. We think that it is better to use more words and sentences to provide a clear presentation of the algorithms.
> >
> > (10) "theoretical contribution is relatively minor"
> >
> > Reply: The IMA method is the main contribution, and it is a heuristic-based method that is not derived from any theory. We use the equilibrium state analysis to provide a theoretical explanation of the method. Our analysis does not need any complex mathematics to explain the trade-off between robustness and accuracy from the perspective of sample margins (see Appendices F, G, and H), which is not a minor contribution. The perspective of equilibrium between classes is new.
> >
> >
> > (11) "the paper does not present the material sufficiently clearly to the reader"
> >
> > Reply: the paper is clear to AnonReviewer3, "The paper is written clearly. There is no difficulty in understanding the content". For readers not familiar with the margin concept, we are sorry that some background was not introduced, e.g. larger margin leading to higher robustness. In Appendix G of the the revised manuscript, we provide the explanation of the basic idea of the IMA method, including the analogy to SVM. The newly added appendices provide more explanations of our method.
> >
> > (12) "experimental evaluation is not sufficiently conclusive in favor of the paper's central hypothesis."
> >
> > Reply: We have tested the IMA method on Moons, MNIST, CIFAR10, SVHN, Fashion-MNIST, and COVID-19 CT. As a comparison, the papers of the other three methods reported experiments on fewer datasets. The idea of increasing margin is shown intuitively on the Moons dataset, and the method performance is quantitatively evaluated on those datasets. With the new experimental results and discussions in Appendices (F, G, and H), we believe the evaluations have become sufficiently conclusive to support the central hypothesis: increasing margins of the samples leads to higher robustness of the model.

---

### Official Review · AnonReviewer3 · 2020-10-25
**The paper proposed an approach to increase the robustness of neural networks for classification tasks. The intuition of the method is to increase the margin of the training samples. The experimental performance of the method is in general on par with the state of the art method.**

**Rating:** 6
**Confidence:** 3

**Review:**

In general, the paper has a good quality. The idea is based on a common intuition that adversarial attacks are most influential to the points close to the decision boundary. The proposed algorithm IMA makes effective use of this intuition and adopts an alternating training process. As an experimental work, the experimental performance of IMA is on par with the state of the art in the experimental settings considered in the paper. This work is important to the ML community. It would be interesting to see further exploration of the algorithm in different testing settings.
The paper is written clearly. There is no difficulty in understanding the content.
Experimental details are provided.

Detailed comments:
1. In (vanilla) adversarial training, the choice of max perturbation $\epsilon_\max$ is usually crucial to the performance of the classifier on noisy and standard data. Is the performance of IMA also that sensitive to the choice of $\epsilon_\max$?
And it is briefly mentioned in section 3.3 that IMA might indicate a good $\epsilon$ for vanilla adversarial training. But this does not say anything about the choice of $\epsilon_\max$ for IMA. And this could be very important to its performance (on clean and noisy data).
2. What might happen to the performance of the method under different choices of $\beta$? It might be interesting to see how IMA deals with the well-known trade-off between robust and standard accuracy, which is currently one of the main concerns of adversarial training methods.

Other cons:
1. Figures are not readable when printed.

Given the above concerns, my initial rating is 6. This may change given further detail of the paper.

---

> ### Author Response · Authors · 2020-11-22
> **Reply to AnonReviewer3**
>
> (1) "In general, the paper has a good quality. The idea is based on a common intuition that adversarial attacks are most influential to the points close to the decision boundary. The proposed algorithm IMA makes effective use of this intuition and adopts an alternating training process. As an experimental work, the experimental performance of IMA is on par with the state of the art in the experimental settings considered in the paper. This work is important to the ML community. It would be interesting to see further exploration of the algorithm in different testing settings. The paper is written clearly. There is no difficulty in understanding the content. Experimental details are provided. "
>
> Reply: we thank the reviewer for the comment and support for our work.
>
> (2) "In (vanilla) adversarial training, the choice of max perturbation is usually crucial to the performance of the classifier on noisy and standard data. Is the performance of IMA also that sensitive to the choice of ε_max? And it is briefly mentioned in section 3.3 that IMA might indicate a good for vanilla adversarial training. But this does not say anything about the choice of  ε_max for IMA. And this could be very important to its performance (on clean and noisy data)."
>
> Reply:  please read appendix F for the choice of  ε_max of IMA. We have added more experimental results and discussions.
>
>
> (3) "What might happen to the performance of the method under different choices of β. It might be interesting to see how IMA deals with the well-known trade-off between robust and standard accuracy, which is currently one of the main concerns of adversarial training methods."
>
> Reply:   please read appendix E for the choice of  β of IMA. We have added more experimental results and discussions.
>
> (4) "Figures are not readable when printed. "
>
> Reply: we are sorry about this: we have to shrink figures to meet the page limit. The figures are in high resolution on computer screen.

---

### Official Review · AnonReviewer2 · 2020-10-25
**Review #2**

**Rating:** 4
**Confidence:** 4

**Review:**

Summary:
The paper proposes increasing-margin adversarial training (IMA) to improve adversarial robustness of a classifier. IMA works by alternating between two algorithms: Algorithm 1 update the model parameters while Algorithm 2 updates the margin estimate. By iteratively increasing the margins from clean training samples, IMA seeks to make the classifier more robust to L-p adversarial perturbations. The authors conducted experiments on the Moons, Fashion-MNIST, SVHN and a CT image dataset to evaluate IMA’s performances against other baselines and found IMA to outperform or be on par with them.

Pro:
+Improving robustness through the margins from clean samples is an interesting approach.

Cons:
-Evaluation on non-standard image datasets used to evaluate adversarial robustness. Lack of evaluation on datasets such as MNIST, CIFAR10/100 or imagenet
-IMA’s assumption that clean samples from different classes are equally spaced from the boundary might not be valid for images. Some classes might require more pixel perturbations to change their ‘ground-truth’ class than others.

Recommendation:
While the idea of improving models’ robustness via increasing margins from clean samples is a refreshing direction to counter adversarial examples, the basis behind the idea of IMA might be flawed. IMA assumes that clean samples from different classes are equally spaced from decision boundaries when in an equilibrium state. However, some classes might require more pixel perturbations to change their ‘ground-truth’ class than others. More discussions and theoretical studies would make IMA more convincing. Another major concern I have is the lack of evaluation on standard image datasets such as MNIST, CIFAR10/100 or imagenet in the paper. Given its current state, I believe the paper is not yet fit for publication.


Comments and Questions:

The results in Fig 6 shows that IMA outperforms other methods but drops sharply at 0.3 noise level to almost match TRADES and adv’s performance, what is its performance vs other methods at levels past 0.3?

The statement “a model robust to noises less than the level of 0.2 is good enough for this application“ is not substantiated by any previous work or experiments.

How is the IMA’s performance against black-box attacks?

---

> ### Author Response · Authors · 2020-11-22
> **Reply to AnonReviewer2**
>
> (1) "Evaluation on non-standard image datasets used to evaluate adversarial robustness. Lack of evaluation on datasets such as MNIST, CIFAR10/100 or imagenet"
>
> Reply: we add additional two experiments on MNIST and CIFAR10 in Appendix D. We do not understand why SVHN and Fashion-MNIST are considered "non-standard".
>
> (2) "IMA’s assumption that clean samples from different classes are equally spaced from the boundary might not be valid for images. Some classes might require more pixel perturbations to change their ‘ground-truth’ class than others….While the idea of improving models’ robustness via increasing margins from clean samples is a refreshing direction to counter adversarial examples, the basis behind the idea of IMA might be flawed. IMA assumes that clean samples from different classes are equally spaced from decision boundaries when in an equilibrium state. However, some classes might require more pixel perturbations to change their ‘ground-truth’ class than others"
>
> Reply:  IMA does not have the assumption that "clean samples from different classes are equally spaced from the boundary". Given the maximum possible sample margin ε_max in IMA , the decision boundary is determined by the clean samples within the distance of ε_max from it . In the equilibrium state, the local densities of noisy samples in different classes are the same along the decision boundary, which does not necessarily mean that the clean samples in different classes are equally spaced from the decision boundary (see Eq.(3) to Eq.(8) in Section 2.3; Eq.(9) and Eq.(10) in Appendix G). We guess the reviewer might get misled by Figure 5 (left), which is only an illustration for a simple scenario: the ε-balls of the samples in two different classes expand and then collide with each other, resulting a local decision boundary that is robust (i.e. far away from the clean samples of the two classes).
>
> IMA indeed will find a decision boundary somewhere in the middle between classes, as shown in the 2D moons dataset.
>
> It is possible that "some classes might require more pixel perturbations to change their ‘ground-truth’ class than others". The problem is we do NOT know the right magnitudes of the pixel perturbations for the samples. Knowing the right magnitudes of the pixel perturbations (i.e. true margins) for the samples is equivalent to knowing the optimal decision boundary. In general, we do NOT have training samples enough to cover the high dimensional input space so that we can do Bayesian classification to get the optimal and robust decision boundary.
>
> What should we do when we do not have enough training samples? In this work, we resort to the basic idea of Support Vector Machine (SVM). For (linear) SVM, if there are only two classes and the data samples are linearly-sparable, then SVM will produce a linear decision boundary in the "middle" between the two classes, and it will have great generalization ability by its theory. The SVM decision boundary is robust: classification output will not change if a small amount of noise δ is added to x as long as the vector norm of δ is smaller than the margin of x. Here, the margin of x is the (minimum) distance between x and the decision boundary.
>
> In general, we do not have enough training samples to do perfect Bayesian classification to find the optimal and robust decision boundary. Therefore, a decision boundary somewhere in the middle between classes is a reasonable and viable choice. (see Appendix H for more discussions)
>
> In general, the data samples in multiple classes are nonlinearly-separable, and the robust decision boundary should be somewhere in the middle between classes, which is the goal that IMA pursues. Because of a nonlinear decision boundary, the margins of the samples are not the same, as shown in Fig. 14 in Appendix F.
>
> From IMA, when an equilibrium state is reached, the distributions (i.e. local densities) of noisy samples in different classes are the same along the decision boundary. This is somewhat analog to Bayesian classification: at the optimal decision boundary, the distributions (densities) of samples in two classes are the same, assuming the classes have equal prior probabilities. From this perspective, the noisy samples, which are generated by IMA, serve as the surrogates of the real samples. Obviously, we cannot claim it is Bayesian classification because noisy samples may not reveal the true distributions.
>
> In some special applications/datasets, if the user knows that the samples in class-1 have significantly larger margins than the samples in class-2, then in the IMA method, the samples in class-1 can be allowed to have significantly larger margins, which can be implemented by using several options: (1) increase β for the samples in class-1, (2) use a larger margin expansion step size for the samples in class-1, and/or (3) use a larger ε_max for the samples in class-1. Clearly, the use case like this would be very rare.

---

> > ### Author Response · Authors · 2020-11-22
> > **continue...**
> >
> > (3) "The results in Fig 6 shows that IMA outperforms other methods but drops sharply at 0.3 noise level to almost match TRADES and adv’s performance, what is its performance vs other methods at levels past 0.3?"
> >
> > Reply: Since it is a synthetic 2D dataset, it is easy to visually identify the optimal decision boundary: a curve almost in the middle between the two classes, and the minimum margin of the samples is about 0.15. When the noise level for the adversarial attack is larger than that, some noisy samples will go across the optimal decision boundary, resulting in miss-classification, which is why the accuracy curve of IMA starts to drop when noise level is larger than 0.15. Obviously, if we use a significantly larger noise level that exceeds the maximum  of the true margins of the samples, the accuracy of every method will approach zero.
> >
> > We note that it is meaningless to defend against very large adversarial noise levels. Noisy samples with adversarial noises can be correctly-classified by humans but may be incorrectly classified by neural networks, which is basically the definition of adversarial noises. From this definition, the true class label of the noisy sample should be the same as the true class label of the clean sample.
> >
> > For this dataset, if the adversarial noise δ is very large, then the noisy sample x+δ can penetrate the true decision boundary (roughly the middle curve), then the true label of the noisy sample will change (but for method evaluation, it is still assumed that the true label of the noisy sample x+δ is the same as the the true label of the clean sample x).  For this dataset, the minimum distance between the two classes is about 0.3. Thus, evaluating any method with adversarial noise level > 0.3 is meaningless on this dataset.
> >
> > (4) "The statement “a model robust to noises less than the level of 0.2 is good enough for this application“ is not substantiated by any previous work or experiments."
> >
> > Reply: the signal to noise ratio of CT imaging for COVID-19 diagnosis should be large enough. For example, in this reference https://www.ncbi.nlm.nih.gov/pmc/articles/PMC7609045/, SNR is 17.0 ± 5.9. When the noise level is 0.2, then SNR is roughly 1/0.2=5 below the required level. When the noise level is 0.1, then SNR is roughly 1/0.1=10, which is reasonable. Basically, if a doctor sees a CT image with noise level larger than 0.2 (see Fig. 10), then the CT imaging machine must have malfunctioned. This kind of noisy images will not be used for diagnosis in clinical practice.
> >
> > (5) "How is the IMA’s performance against black-box attacks?"
> >
> > Reply: In Appendix D, we add additional two experiments on MNIST and CIFAR10, using PGD white-box attack and SPSA black-box attack. The results show that IMA does not do gradient obfuscation.

---

### Official Review · AnonReviewer4 · 2020-10-28
**Theoretically working but not significant with real data**

**Rating:** 4
**Confidence:** 4

**Review:**

The authors propose a new training method, named Increasing Margin Adversarial (IMA) training, to improve DNN robustness against adversarial noises. The IMA method increases the margins of training samples by moving the decision boundaries of the DNN model far away from the training samples to improve robustness. Under strong 100-PGD whitebox adversarial attacks, the authors evaluated the IMA method on four publicly available datasets.

Overall, I vote for ok but not goor enough - rejection. The proposed strategy sounds reasonable and worked well with simple dataset, the Moons dataset. However, when it was applied to more complicated real dataset such as Fashion-MINST, SVHN, and COVID-19 CT image dataset; there was no significant achievement if compare to the MMA approaches. Thus further investigation is needed to convince benefit of the IMA on real datasets.

In addition, the authors tested only one medical image dataset, COVID-19 CT image dataset. Since there are multiple modalities in the medical field and the diversity among datasets are quite large, it is too early to emphasize the advantage of the proposed method in the medical field in general like the last phrase in the conclusion “We hope our apporach may facilitate the development of robust DNNs, especially in the medical field.”

---

> ### Author Response · Authors · 2020-11-22
> **Reply to AnonReviewer4**
>
> (1) "The proposed strategy sounds reasonable and worked well with simple dataset, the Moons dataset. However, when it was applied to more complicated real dataset such as Fashion-MINST, SVHN, and COVID-19 CT image dataset; there was no significant achievement if compare to the MMA approaches. Thus further investigation is needed to convince benefit of the IMA on real datasets."
>
> Reply: the benefit of the IMA is discussed in Appendices (F, G, and H) with more experimental results.
>
> We note that the comment from AnonReviewer3 highlights our contribution, "The idea is based on a common intuition that adversarial attacks are most influential to the points close to the decision boundary. The proposed algorithm IMA makes effective use of this intuition and adopts an alternating training process. As an experimental work, the experimental performance of IMA is on par with the state of the art in the experimental settings considered in the paper. This work is important to the ML community."
>
> Also, please read "Summary of the Revision" posted on this forum.
>
> (2) "In addition, the authors tested only one medical image dataset, COVID-19 CT image dataset. Since there are multiple modalities in the medical field and the diversity among datasets are quite large, it is too early to emphasize the advantage of the proposed method in the medical field in general like the last phrase in the conclusion “We hope our apporach may facilitate the development of robust DNNs, especially in the medical field.”
>
> Reply: we changed the sentence to "We hope our approach may facilitate the development of robust DNN applications, especially for COVID-19 diagnosis using CT images.", which is more specific. We feel that we need to do something for the COVID-19 situation. In many countries, CT imaging is used as the primary diagnostic tool(https://ieeexplore.ieee.org/document/9069255)

---

### Author Response · Authors · 2020-11-22
**Summary of the Revision**

We thank the four reviewers for spending your valuable time reading our manuscript. Since we received the comments, we have been working day and night to do new experiments and analyze the results in order to answer the questions from the reviewers. Here, we provide a summary of the revision, and please read the revised paper for the details.

Please note that "a lack of state-of-the-art results does not by itself constitute grounds for rejection. Submissions bring value to the ICLR community when they convincingly demonstrate new, relevant, impactful knowledge. Submissions can achieve this without achieving state-of-the-art results." https://iclr.cc/Conferences/2021/ReviewerGuide#faq

What is new in the paper?

IMA algorithms, theoretical analysis and experiments are all new.

What is the relevant and impactful knowledge in the paper?

The comment from AnonReviewer3 can well answer this question, "The idea is based on a common intuition that adversarial attacks are most influential to the points close to the decision boundary. The proposed algorithm IMA makes effective use of this intuition and adopts an alternating training process. As an experimental work, the experimental performance of IMA is on par with the state of the art in the experimental settings considered in the paper. This work is important to the ML community."

What is new in the revised paper?

(1) In Appendix D, we add two additional experiments on MNIST and CIFAR10.

(2) In Appendix E, we show that how β can be used to make a trade-off between robustness and accuracy, and we also show that the trade-off is highly nonlinear: a small decrease in accuracy on clean data can result in a large increase in accuracy on noisy data. The nonlinear trade-off between robustness and accuracy makes it difficult to directly compare two methods (e.g. IMA and MMA), as different methods make different trade-off between robustness and accuracy.

(3) In Appendix F, we discuss how the allowed maximum margin (i.e. ε_max) may affect the performance and how to choose its value. The IMA method has a clear explanation of the trade-off between robustness and accuracy, and it enables an easy way for the user of our method to choose the parameter ε_max, which is difficult to do for other methods.

(4) In Appendix F, we also show that for the COVID-19 application, vanilla adversarial training with ε =0.15 achieved the best performance, compared to other methods. The advantage of vanilla adversarial training is that it is simple to implement and has relatively low computational cost, but it is not easy to find a good ε. The margin distribution estimated by IMA can reveal a good range of the 𝜀.

(5) In Appendix G, we provide further explanation of the IMA method.

(6) In Appendix H, we discuss the trade-off between robustness and accuracy

What are the benefits of IMA?

(1) For algorithm designers/researchers: the IMA method and the experimental results explain the trade-off between robustness and accuracy from the perspective of sample margins (see Appendices F and G). Our work has demonstrated that "common intuition that adversarial attacks are most influential to the points close to the decision boundary" can be materialized into algorithms to improve adversarial robustness, which points out a promising direction for defense against adversarial attacks.

(2) For the user of IMA to make robust applications (e.g. COVID-19 CT): the IMA method provides a convenient and efficient way to make a trade-off between robustness and accuracy (see Appendices F and G).

---

> ### Author Response · Authors · 2020-11-25
> **update**
>
> Dear reviewers,
>
> We updated the paper again on the last day to include more experimental results and discussions. Please read the lasted version that has 15 Figures and 21 Tables showing results on 6 datasets. Thank you for your valuable comments.
>
> Best,
>
> Authors

---

### Decision · Program_Chairs · 2021-01-07
**Final Decision**

**Decision:**

Reject

**Comment:**

The paper proposes a margin-based adversarial training procedure. The paper is lacking in terms of proper dicussion of related literature e.g. similarity and differences to MMA, the "theoretical" discussion on page 5 is incomplete as there is no way how one can estimate the perturbed samples to do the analysis (the authors seem to implicitly already assume that the adversarial samples lie on the decision boundary) and the underlying assumptions are not clearly stated, the reported robust accuracies
(see https://github.com/fra31/auto-attack for a leaderboard of adversarial defenses) on MNIST and CIFAR10 are worse than that of MMA which are in turn worse than SOTA. Thus this paper is below the bar for ICLR.